# Goal-directed navigation in humans and deep reinforcement learning agents relies on an adaptive mix of vector-based and transition-based strategies

Denis C. L. Lan◉*, Laurence T. Hunt◉, Christopher Summerfield◉

Department of Experimental Psychology, Medical Sciences Division, University of Oxford, Oxford, United Kingdom

◉ These authors contributed equally to this work.
* denis.lan@psy.ox.ac.uk

## Abstract

Much has been learned about the cognitive and neural mechanisms by which humans and other animals navigate to reach their goals. However, most studies have involved a single, well-learned environment. By contrast, real-world wayfinding often occurs in unfamiliar settings, requiring people to combine memories of landmark locations with on-the-fly information about transitions between adjacent states. Here, we studied the strategies that support human navigation in wholly novel environments. We found that during goal-directed navigation, people use a mix of strategies, adaptively deploying both associations between proximal states (state transitions) and directions between distal landmarks (vectors) at stereotyped points on a journey. Deep neural networks meta-trained with reinforcement learning to find the shortest path to goal exhibited near-identical strategies, and in doing so, developed units specialized for the implementation of vector- and state transition-based strategies. These units exhibited response patterns and representational geometries that resemble those previously found in mammalian navigational systems. Overall, our results suggest that effective navigation in novel environments relies on an adaptive mix of state transition- and vector-based strategies, supported by different modes of representing the environment in the brain.

## Introduction

Humans and other animals are remarkably flexibly at planning and navigation, leading neuroscientists as early as Tolman to suggest that we possess 'cognitive maps'—mental representations of environmental structure that tell us how regions of physical space relate to each other [1]. Neuroscientists have long believed that cognitive maps are important for goal-directed navigation [2], accounting for flexible behaviors

**Data availability statement:** All behavioral data files are available from OSF (doi.org/10.17605/OSF.IO/W39D5). Code for this project is available from https://github.com/denis-lan/navigation-strategies (archived at doi.org/10.5281/zenodo.15741947).

**Funding:** This research is supported by a European Research Council Consolidator Grant (https://erc.europa.eu/apply-grant/consolidator-grant) (725937 - CQR01290.CQ001) and a Wellcome Trust Discovery Award (https://wellcome.org/grant-funding/schemes/discovery-awards) (227928/Z/23/Z) to C.S. and a Sir Henry Dale Fellowship from the Royal Society and the Wellcome Trust to L.T.H. (https://royalsociety.org/grants/henry-dale/) (208789/Z/17/Z). This research is supported by the Social Science Research Council (Singapore) and administered by the Ministry of Education, Singapore, under its Social Science Research Council Graduate Research Fellowship (https://www.ssrc.edu.sg/initiatives/grf/overview/) ([SSRC-2024-001]) to D.C.L.L. Any opinions, findings and conclusions or recommendations expressed in this material are those of the authors and do not reflect the views of the Social Science Research Council (Singapore) and the Ministry of Education, Singapore. D.C.L.L. is also supported by the Clarendon Fund and the Christopher Welch Trust, and L.T.H. is also supported by a strategic Longer and Larger Grant from the BBSRC (https://gtr.ukri.org/projects?ref=BB/W003392/1) (BB/W003392/1). The funders played no role in study design, data collection and analysis, decision to publish, or the preparation of the manuscript.

**Competing interests:** The authors have declared that no competing interests exist.

**Abbreviations:** PCA, principal component analysis; PPO, proximal policy optimization; RL, reinforcement learning.

such as navigating around obstacles or finding shortcuts through unexplored space [1,3,4]. In humans, cognitive maps may also support multi-step reasoning processes that permit problem solving in nonspatial domains [5–8]. However, we still do not have a complete account of how a cognitive map might be used for goal-directed planning and inference.

Proposed answers to this puzzle often make varying assumptions about the format of the cognitive map [8,9]. One potential format is topological. Topological representations encode which states are linked to which, without making strong claims about how the map is spatially organized. There is good evidence that the brain learns associations between states that are experienced in succession, allowing it to compute the likelihood that a successor will be attained from the currently occupied state [10,11]. Such a representation could be used to rapidly infer which local states might bring the agent closer to a distal goal or reward [12], especially in cluttered environments [13,14]. As goals are learned, animals may come to jointly represent current and intended states (such as locations within an environment) in a single cell or neural population, allowing them to become linked in memory [15–18]. Alternatively, offline replay [19] or explicit rehearsal [20] might be used to mentally explore a cognitive map, to identify the most promising path to reach a destination [19,21,22]. These theories assume that cognitive maps are topological representations of the world, which encode transitions between states in a graph-like structure that is not necessarily constrained to a particular geometry [7].

An alternative format for the cognitive map is geometrical. Under this assumption, goal-directed behaviors require agents to represent the world on a low-dimensional Euclidean manifold, like a physical map. These representations allow states to be organized in a well-defined coordinate system that encodes not just the associative strength but also the distance and angle (vector) between states. This allows animals to infer the movement direction required to reach a goal state from any given location [23,24], which is particularly helpful for inferring routes between any two points in space without prior navigational experience [3,25]. In one interesting model, a neural network trained to predict the next state (by learning conjunctive representations of objects and locations) is able to infer how each state in a cognitive map relates to every other, including transitions it has not experienced [26]. It is believed that in the mammalian navigation system, state (or place) codes in the hippocampus and entorhinal cortex are anchored to a Euclidean coordinate system defined by the periodic activity of grid cells [24,27]. Consistent with this observation, the model learned grid-like units with regularly spaced firing patterns [26]. Similar units have been obtained in deep networks which learned to take shortcuts in a 3D environment to maximize reward [25]. This may form a substrate for geometric forms of inference in both spatial and nonspatial domains [24,28].

These two forms of representation need not be mutually exclusive. Flexible navigation likely requires agents to use maps in both formats [9,29]: associative, 'state transition-based' strategies allow us to navigate around obstacles in well-learnt, cluttered environments [13], while spatial, 'vector-based' strategies allow us to infer new routes and shortcuts through unknown regions of space, especially in open-field

environments [3,25]. However, we know little about how humans or other animals may combine these strategies during navigation. This is partially because existing research has focused on planning problems where participants have extensive knowledge or experience of the environment's transition structure [30, 31], where arbitration between different strategies might be less important. However, humans are adept at planning even with minimal knowledge of this structure—for example, when visiting a new city, we plan routes successfully despite sparse knowledge of the city's layout by anchoring our navigation to a few key landmarks [32,33]. To achieve this, we recycle navigational strategies that have been used in other, comparable situations, to learn about novel environments "on the fly" while navigating. Navigating in unfamiliar environments can thus be construed as a 'meta-learning' problem, where agents are tasked with acquiring strategies that will help them learn about novel task environments more efficiently [34,35].

Here, we studied how humans arbitrate between these two complementary 'transition-' and 'vector'-based strategies for 'few-shot' goal-directed navigation. Recreating a commonly experienced navigational problem, we assumed that people have some rough sense of where the goal and some landmarks lie (e.g., from glancing at a map before setting off) but otherwise find their way by naïve exploration. To achieve this, we used a stylized task in which participants were first given a bird's eye view of a grid world, in which each square was identified by a unique hidden object. They were prompted to click on grid squares to reveal landmark and goal objects (the 'map reading' phase). Subsequently, participants attempted to navigate to the goal by choosing either a direction to go in, or a state to transition to (the 'navigation' phase). Unlike most previous studies, we presented participants with a new navigation problem on each trial, with a new layout, and new markers for states denoting the goal, landmarks, and standard locations. Our main question was the nature of the navigational policy that people have meta-learned: how they combined transition-based strategies and vector-based strategies to solve these novel navigation problems.

We also trained deep neural networks with reinforcement learning (RL) to perform an equivalent task. As with humans, we trained the networks in a regime where each trial involved a unique environment. This required them to "meta-learn" a strategy for few-shot navigation that involved either directions, or states, or both. We found striking similarities in the meta-learnt policies that humans and deep RL agents used to solve the task, suggesting these strategies were well adapted for 'few-shot' navigation.

## Results

Participants (*n* = 401 over 3 experiments) performed a computerized task that involved navigating an 8 × 8 grid. Every grid square contained either a traversable object or an obstacle (Fig 1A). In the *map reading* phase of each trial, participants saw a bird's eye view of the grid (Fig 1B) and were forced to click on a sequence of single blue squares, each of which briefly revealed a 'landmark' object for 3 s (or until the next square was clicked). There were 16 such landmark exposures, distributed across 2–16 unique landmark locations. They then clicked on a single yellow square, which revealed the goal object for this trial.

Each trial's map reading phase was followed by a *navigation* phase, during which they moved between objects which were displayed singly and centrally, attempting to find the goal object in as few steps as possible (Fig 1C). Participants started at a random location that they had not learnt about in the map-reading phase that was four steps away from the goal. On each step, participants could choose to navigate the grid in one of two ways: they could either choose a *direction* to go in (up, down, left, or right arrows) or a *state* to transition to (the immediately adjacent objects located north, south, east, and west, presented in random order). Crucially, while both response methods allowed participants to move to the same set of adjacent states, participants' choice of response method allowed us to infer whether they were likely using a vector-based (focusing on *directions*) or a transition-based (focusing on *states*) strategy. Note that our use of 'transitions' in this paper refers specifically to computations involving transitions between states and not actions or directions. If participants were adjacent to a wall or an obstacle, an image of a boulder would be shown as one of the adjacent states, and clicking on the boulder, or on an arrow in the direction of the obstacle, would have no effect. Every trial (*n* = 32

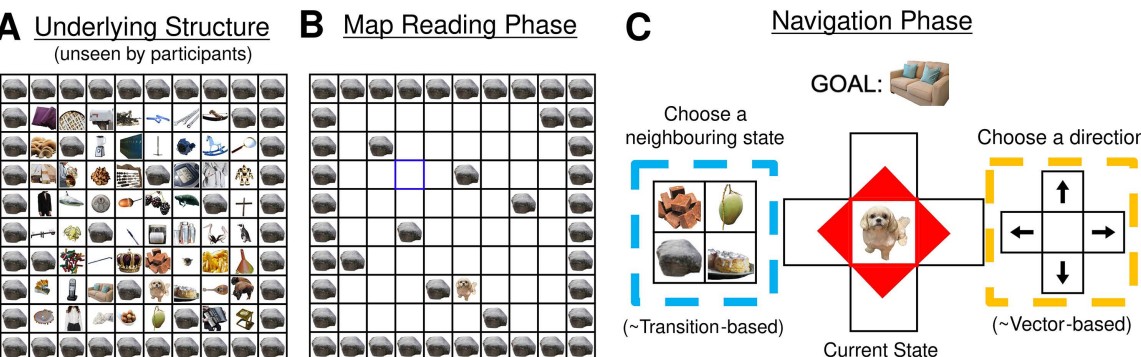

**Fig 1. Task design and experimental set-up. A:** underlying structure of the 8 × 8 grid, unseen by participants. Every state is represented by an image of an object, and these objects and their positions change on every trial. **B:** schematic diagram of the 'map reading' phase of each trial. Participants see a top–down view of the grid with objects obscured and successively click on blue squares to reveal 'landmark' objects at the location. After 16 clicks have been completed, a yellow square appears. Clicking on the yellow square reveals the 'goal' object for the trial. **C:** schematic diagram of the navigation phase of each trial. Participants start in a random, previously unobserved location and are tasked with navigating to the 'goal' object they had just learnt about (displayed at the top). They can navigate in two ways. First, they could choose a *direction* to travel in by clicking on the corresponding arrow (highlighted yellow). This is analogous to using a 'vector-based' strategy. Alternatively, they could choose an adjacent *state* to travel to by clicking on one of the associated images (displayed in a random order; highlighted blue). This corresponds to using a 'transition-based' navigation strategy. Both response methods were equivalent in that they both only allowed participants to move to the four adjacent states, but setting up the response methods in this way allowed us to determine if participants were focusing more on the direction they were travelling in, or the identity of the next state they would be transitioning to.

in Experiment 1, $n = 24$ in Experiments 2 and 3) involved different images of objects, landmark locations, and obstacle locations, requiring participants to plan 'few-shot', i.e., with minimal information about the environment, and to adjust their wayfinding on the fly.

We performed three variants of this experiment. Experiment 1 was designed to determine the relative importance of vector- and transition-based strategies in few-shot goal-directed navigation. Experiment 2 was a pre-registered experiment that sought to confirm our hypotheses about how participants arbitrated between the two strategies. Experiment 3 extends our behavioral findings by allowing people to freely sample landmarks of their choice, and studying how this impacts goal-directed navigation.

### Experiment 1: Navigation is most effective when adaptively selecting between vector- and transition-based strategies

In Experiment 1 ($n = 200$), we tested the hypothesis that participants would navigate to the goal more efficiently if they could freely choose when to deploy a transition-based strategy (i.e., choose among states) or a vector-based strategy (i.e., choose among directions). We varied the task over four conditions in a repeated measures design, where they could (1) freely choose which strategy to use on each turn ('*both*' condition), (2) were obliged to use directions ('*directions only*' condition), (3) were obliged to use states ('*states only*' condition), and (4) where they were forced to alternate between the two strategies at random ('*random alternation*' condition). This within-subjects factor was crossed with another variable that determined whether participants navigated in (1) a cluttered grid with intervening obstacles, or (2) an open-field environment, giving a 4 × 2 (mixed) *strategy × environment* factorial design.

Fig 2A shows participants' number of steps to goal on the four strategy conditions in both open field and cluttered versions of the task. Participants reached the goal most efficiently in the '*both*' condition, where they could always freely choose between either strategy, compared to all other conditions. A linear mixed effects model with trial number, strategy, and environment as predictors of log-transformed number of steps revealed that overall, compared to the *both* condition,

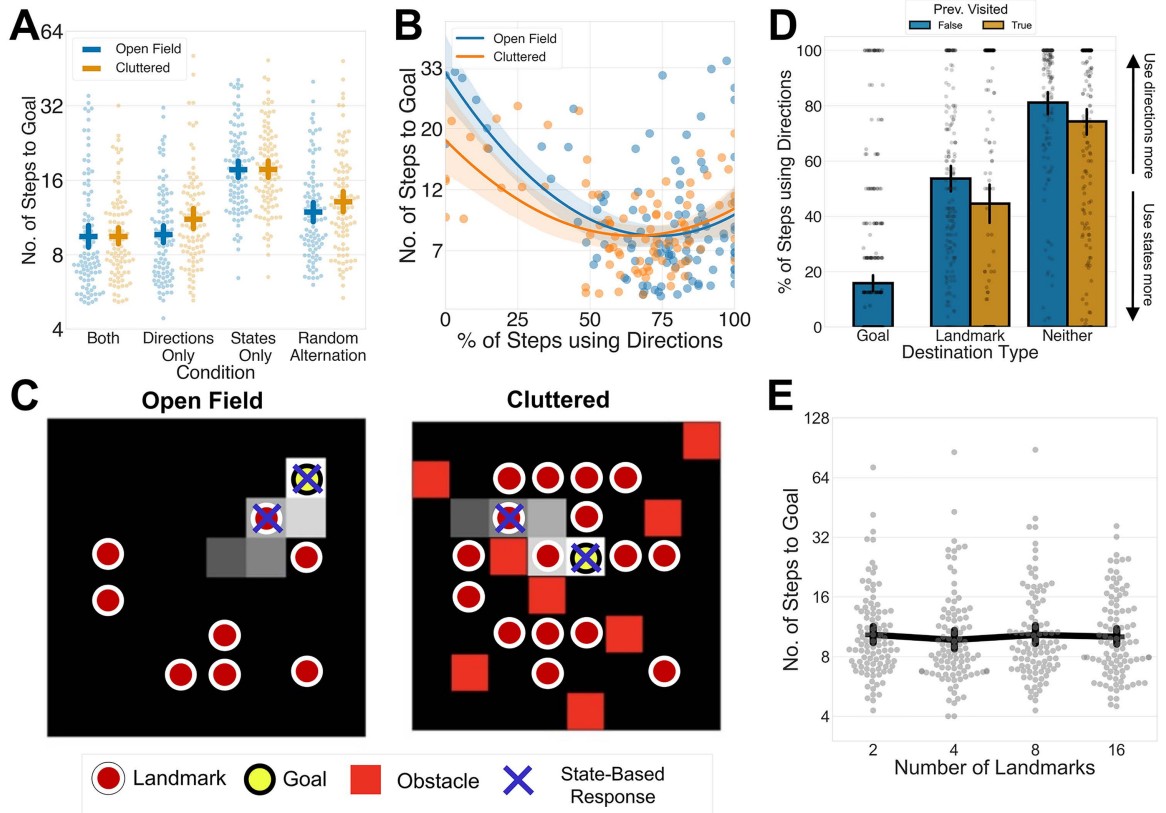

**Fig 2. Human participants benefit from freely arbitrating between vector- and transition-based strategies. A:** Performance for each participant across the different conditions of Experiment 1. *Y*-axis represents number of steps taken to reach a goal on a logarithmic scale, while *x*-axis represents the different conditions. Each dot represents an individual participants' performance on each condition, and the dashes represent the mean performance across all participants, with error bars representing the 95% CI. **B:** Relationship between the proportion of steps made using *direction* responses (*x*-axis) and the number of steps taken to reach a goal (*y*-axis; represented on a logarithmic scale) for each participant. The different colors represent different types of environments, while the lines represent the best-fitting quadratic curve. **C:** Selected sample participant trajectories on the task. Participants' trajectories progress from the darker squares to the lighter squares. A red circle indicates the location of a landmark, and a red square indicates an obstacle in the cluttered condition. A yellow square indicates the goal for the trial. A cross indicates when participants used a *state* response to get to the state. In these trajectories, participants use *direction* responses most of the time but use *state* responses to get to a landmark or goal. **D:** Participants' use of *direction* responses (*y*-axis) as a function of destination type (i.e., goal, landmark, or non-landmark; *x*-axis) and whether the state had been visited before (color of bar). Error bars represent the 95% CI. **E:** Performance for each participant across the different numbers of landmarks in Experiment 2. *Y*-axis represents number of steps taken to reach a goal on a logarithmic scale, while *x*-axis represents the different numbers of landmarks. Each dot represents an individual participants' performance on each condition, and the dashes represent the mean performance across all participants, with error bars representing the 95% CI. Data and code underlying this figure are available at https://osf.io/w39d5/ and https://github.com/denis-lan/navigation-strategies, respectively.

participants took more steps to reach the goal in the other three conditions: (linear mixed effects model: *both* versus *directions*, *β* = 0.098, *SE* = 0.028, *z* = 3.48, *p* < .001, *both* versus *states*, *β* = 0.63, *SE* = 0.033, *z* = 19.17, *p* < .001; *both* versus *random*, *β* = 0.28, *SE* = 0.030, *z* = 9.32, *p* < .001). The *states* only condition saw the worst performance, followed by the *random alternation* and then the *directions* condition. This pattern of results confirms our hypothesis that few-shot navigation relies on a combination of both vector- and transition-based strategies.

In the same mixed effects model, we also observed a significant interaction between *directions* and environment type (*β* = 0.14, *SE* = 0.06, *z* = 2.53, *p* = .01). This implies that deficit in performance on the *directions* condition was more apparent in the cluttered than the open field environment. This is likely because open-field environments are more amenable to

vector-based strategies as it always optimal to follow the vector to the goal, while cluttered environments require agents to also use transition-based strategies to successfully navigate around obstacles [13,14,36,37].

This finding implies that both vector- and transition-based strategies are important for few-shot navigation. We might thus expect that when participants are free to choose between both strategies, those who strike a balance between the two would perform better than those who predominantly relied on either directions or states. Focusing on the '*both*' condition, we thus performed a linear regression model predicting each participant's mean log-transformed number of steps with linear and quadratic terms for the proportion of steps a participant made using directions and states, environment type, and their interactions. As shown in Fig 2B, this revealed a quadratic relationship between performance and proportion of 'direction' steps that was independent of environment type ($\beta = 14.50$, $SE = 5.29$, $t(190) = 2.74$, $p = .007$). Participants that tended to rely exclusively on one strategy or the other performed more poorly. This quadratic relationship replicated in all subsequent versions of the task, including as a pre-registered hypothesis in Experiment 2 (S3A Fig).

Next, again focusing on the '*both*' condition, we asked how participants chose between strategies over the course of a trial. Fig 2C shows sample trajectories in open-field and cluttered environments. Anecdotally, we observed that participants tended to rely mostly on directions to head towards the goal but relied on transitions to fine-tune their navigation when near to landmarks (which they recognized from the map-reading phase). Quantitative analysis confirmed this finding. Participants preferred vector-based navigation strategies overall; they chose directions on 73.8% ($SD = 23.1$%) of steps in the open field environment, and 63.6% ($SD = 26.1$%) in the cluttered environment (this difference being significant, $t(194) = 2.88$, $p = 0.004$). However, when adjacent to goals or landmarks, participants were more prone to choose states (Fig 2D). A mixed effects logistic regression revealed that across both environment types, state-based responses were more likely to be chosen to reach a goal ($\beta = 4.20$, $SE = 0.21$, $z = 19.45$, $p < .0001$) or landmark ($\beta = 2.17$, $SE = 0.16$, $z = 13.48$, $p < .0001$). Overall, this pattern of results suggests that humans were broadly relying on vector-based strategies to head in the general goal direction but using transition-based strategies proximal to landmarks and goals to 'fine-tune' their navigation.

Just like a hiker with a map, participants could jointly rely on cues from map-reading and from the organization of locations already encountered on their trajectory. We thus compared strategy use at locations defined by objects that had been encountered before and found that state-based responses were also more popular at previously encountered objects (Fig 2D; $\beta = 0.45$, $SE = 0.10$, $z = 4.37$, $p < .0001$). This effect of previous encounter was even stronger for non-landmarks than for landmarks (landmark × previously visited interaction: $\beta = -1.21$, $SE = 0.18$, $z = -6.84$, $p < .0001$). In other words, participants were exploiting information about the states that they have picked up over the course of their navigation trajectories and using it to estimate the goal direction.

Overall, our results suggest that combining vector- and transition-based strategies is crucial for good few-shot navigation performance, perhaps by using vector-based strategies to head in the general goal direction but tweaking navigation with transition-based strategies near landmarks and other familiar locations.

## Experiment 2: number of landmarks during map reading does not impact navigation performance

The main goal of Experiment 2 was to replicate the major effects described above in a pre-registered study [38]. Because many of the most interesting findings above were observed when both directions and states were available, we asked our new cohort ($n = 100$) to perform exclusively the '*both*' condition in a cluttered environment. The results confirmed our main pre-registered hypothesis that participants prefer direction-based responses in general but use state responses more at landmark and goal states. In additional exploratory analyses, we also replicated the finding described above that participants use state-based responses more to reach states that they had previously encountered during navigation (S3B Fig).

Having participants complete more trials in the '*both*' condition also afforded us the opportunity to study how the number of landmarks selected during the map reading phase affected navigation performance when participants were allowed to freely arbitrate between response methods. Participants each performed four blocks with 2, 4, 8 or 16 landmarks being

highlighted in the map reading phase. However, we kept the number of exposures constant, so that participants viewed 2 landmarks 8 times each, 4 landmarks 4 times each, 8 landmarks 2 times each, or 16 landmarks just once in different blocks. We conjectured that there might be a U-shaped relationship between the number of landmarks and navigation performance, where an attempt to encode too many landmarks or too few would both be detrimental during the map reading phase.

Surprisingly, the data revealed no significant relationship between the number of landmarks and performance (Fig 2E). In our pre-registered linear mixed effects model, there were no significant differences in log-transformed steps-to-goal between the four landmark number conditions. To determine whether there was evidence in favor of the null hypothesis that there was no relationship between the number of landmarks and performance, we performed a Bayesian model comparison between two models: a mixed effects linear model with trial number and number of landmarks as explanatory variables, and a second model that excluded the number of landmarks as a variable. The log Bayes Factor for the model with number of landmarks over the model without number of landmarks was −27.39, indicating very strong evidence in favor of the null hypothesis that there was no association between number of landmarks and log-transformed number of steps to goal. This suggests that participants were equally good at planning regardless of whether they had knowledge of few but strongly encoded, or more but weakly encoded landmarks.

The lack of effect of the number of landmarks was likely due to a trade-off between having more landmarks throughout the environment and being able to remember each individual landmark well. Indeed, participants were naturally more likely to encounter a landmark early when there were more landmarks, but participants' navigational accuracy improved more after encountering landmarks when there were fewer landmarks to remember (S5 Fig).

## Deep meta-reinforcement learning models learn qualitatively similar behavioral policies to humans

Our results so far suggest that participants navigate to goals using a combination of strategies. Specifically, they have a general tendency to use a vector-based strategy but switch to a transition-based strategy when adjacent to landmarks and goals. However, this descriptive finding does not tell us anything about the normative status of this policy – is this what participants should be doing to maximize navigational efficiency? It could be, for example, that this pattern of behavior reflects an unrelated bias, such as a preference to choose familiar states when available, but does nothing to help reach the goal. To address this question, we meta-trained a deep neural network to solve the same spatial navigation task and compared its behavior to that of our human participants.

Specifically, we trained deep reinforcement learning agents (RL agents) to perform a task analogous to Experiment 1, with all four conditions (i.e., *both*, *directions*, *states*, and *random alternation*) and types of environments (i.e., open field or cluttered) interleaved. To recreate the memory constraints on landmark encoding that humans experience, we assumed that RL agents possessed noisy estimates of 'landmark' and 'goal' locations that they recalled when the corresponding image appeared on the screen (see "Materials and methods"). Like humans, RL agents were required to navigate on completely new maps (with different 'object' locations, landmarks, and obstacles) on every trial. Hence, agents were forced to meta-learn a policy for few-shot planning that allowed them to navigate on maps on which they had minimal knowledge and no prior navigational experience. The agent was trained using Proximal Policy Optimization (PPO) and consisted of an LSTM with separate policy and value heads (Fig 3A). We trained 20 iterations of each RL agent, each with a different initialization seed, to ensure replicability across the neural network models. The performance of the RL agents improved throughout training and had plateaued by the end of the training run (S7 Fig). We then tested the agents on new trials of the same task with its weights frozen, forcing it to exploit its meta-learnt policy for navigating in new environments.

Like humans, RL agent performance on the task was significantly worse when it was not allowed to freely arbitrate between directions and states (Fig 3B), supporting the idea that a combination of vector-based and transition-based strategies are necessary for effective few-shot navigation. We conducted statistical analyses on the agents' behavior in the same way as for humans, bearing in mind that the sources of intrinsic variability were different (relating exclusively to

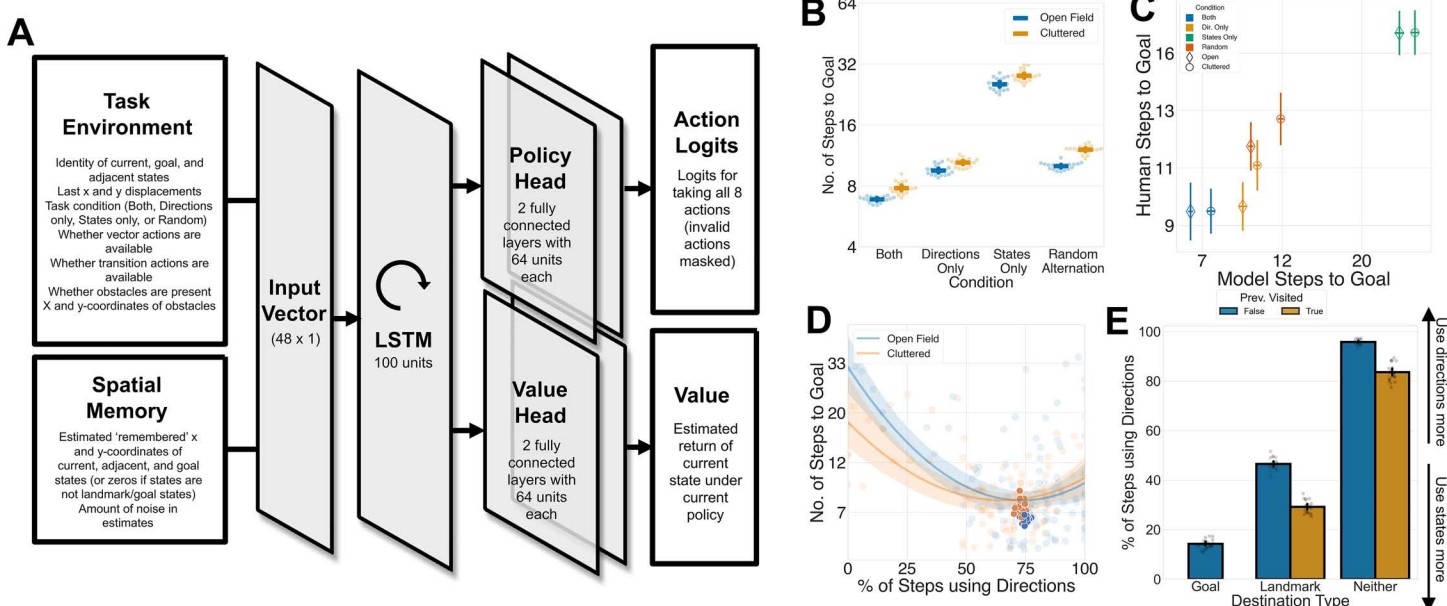

**Fig 3. Deep RL model meta-trained for few-shot navigation recapitulates key features of human behavior. A:** architecture of the deep reinforcement learning network. The network consisted of an LSTM with separate policy and value heads. **B:** model performance across the different conditions. *Y*-axis represents number of steps taken to goal on a logarithmic scale, while *x*-axis represents the different conditions. Each dot represents an individual model's performance on each condition, and the dashes represent the mean performance across all participants, with error bars representing the 95% CI. **C:** scatterplot showing the correspondence between model performance (*x*-axis, as measured by number of steps to goal, represented on a logarithmic scale) and human performance (*y*-axis) for each condition. The colors of the scatter points represent different action conditions, while the shapes represent the type of environment. Error bars represent the 95% confidence interval. **D:** Performance (as measured by number of steps to goal; on *y*-axis, on a logarithmic scale) and proportion of steps using vectors (*x*-axis) of the models (colored dots) superimposed on scatter plot of human performance (translucent dots) and best-fit quadratic curve for the relationship between the proportion of steps made using vector-based responses and performance in humans. **D:** Models' use of vector-based responses (*y*-axis) as a function of destination type (i.e., goal, landmark, or non-landmark; *x*-axis) and whether the state had been visited before (color of bar). Each dot represents the behavior of an individual model, with error bars representing the 95% CI. Data and code underlying this figure are available at https://osf.io/w39d5/ and https://github.com/denis-lan/navigation-strategies, respectively.

the random initial seed and the structure of the map experienced) and that the larger quantity of data rendered our results highly significant.

The patterns of data were very similar between humans and RL agents (Fig 3C). Compared to the '*both*' condition, models also took more steps to reach the goal in the '*directions only*', (linear mixed effects model; $\beta = 0.31$, $SE = 0.01$, $z = 28.45$, p < .0001), '*states only*', ($\beta = 1.29$, $SE = 0.01$, $z = 99.84$, $p < .0001$) and *random alternation* conditions ($\beta = 0.41$, $SE = 0.01$, $z = 6.67$, $p < .0001$). Unlike humans, there was a main effect of obstacles on performance ($\beta = 0.13$, $SE = 0.01$, $z = 7.42$, $p < .0001$), while the interaction of the *directions* condition and the presence of obstacles appeared to be in the opposite direction ($\beta = -0.04$, $SE = 0.02$, $z = -2.07$, $p = 0.04$). Notably, when allowed to freely arbitrate between both strategies, models learnt a balance between strategies that was close to that of the median human ($M_{human} = 73.8\%$, $M_{model} = 73.2\%$) and near the minimum point in the quadratic relationship between use of vectors and steps-to-goal in humans (Fig 3D).

Moreover, the behavioral policy that the models learnt for arbitrating between the two strategies were strikingly similar to that of humans (Fig 3E). Like humans, models were more likely to use directions overall (mixed effects logistic regression; intercept: $\beta = -2.15$, $SE = 0.10$, $z = -19.97$, $p < .0001$), but favored states when adjacent to goals ($\beta = 3.66$, $SE = 0.13$, $z = 28.58$, $p < .0001$) and landmarks ($\beta = 2.38$, $SE = 0.12$, $z = 20.68$, $p < .0001$). Like humans, this was also modulated by

whether the states had been previously visited: models were also more likely to use state-based responses to get states that had been previously visited during navigation ($\beta = 1.30$, $SE = 0.10$, $z = 12.42$, $p < .0001$), likely exploiting knowledge about the transition structure that they had learned during online navigation. Like in humans, this effect was stronger for non-landmark states that models had no prior knowledge about than for landmarks (previously visited × landmark interaction: $\beta = -0.92$, $SE = 0.14$, $z = -12.34$, $p < .001$). Additionally, like humans, RL agents were also unaffected by the number of landmarks (S8A Fig). Thus, RL agents, which have been optimized to maximize reward and thus to reach the goal in the fewest possible steps, replicated almost all the key signatures of human navigation strategies. Assuming that agents have converged on near-optimal policies, this suggests that the strategies adopted by humans were also roughly optimal for this task, under the assumptions built into the deep RL agents.

**Deep meta-reinforcement learning models spontaneously develop modules for 'vector-' and 'transition-based' strategies**

We found that RL agents learned human-like strategies for arbitrating between vector- and transition-based strategies. Thus, we next examined the representations that they learned to better understand how transition- and vector-based strategies may be implemented computationally. We hypothesized that models would learn to perform computations for implementing navigational strategies using their recurrent units, which are potentially analogous to recurrent hippocampal-neocortical interactions in humans and other animals.

In mammalian brains, navigation is thought to be supported by cells in the hippocampus and neocortex that have characteristic response patterns to different aspects of space and objects. This includes cells that respond reliably to specific spatial locations [39,40], the presence of landmarks and other perceptual stimuli [41–43], or a conjunction of both (e.g., a perceptual stimulus in a specific region of space) [44–46]. To determine whether units in our deep RL agents exhibited similar response patterns, we studied the activations of the LSTM units as agents navigated four different environments with different landmark configurations. For each unit, we asked whether its mean activation in each grid state was significantly modulated by (1) which quadrant of space the grid state was located, (2) whether the grid state was on or adjacent to a landmark, or (3) an interaction of these variables. We show examples of cells modulated in each of these 3 ways in Fig 4A, and the overall fractions of neurons modulated by each variable in Fig 4B (top).

We found that many units (22.9%) responded consistently to the same region of space across distinct environments (Fig 4A, top). These units were largely invariant to the location of environmental landmarks. These cells often showed peak firing rates near the boundaries of the environment, thus resembling boundary cells in the mammalian entorhinal cortex [39] and subiculum [47]. Alternatively, these cells also resemble neurons in the rodent hippocampus that respond to location but show no or limited global remapping [48–50]. However, a distinct population of units (6.4%) fired when the RL agent was adjacent to a landmark, irrespective of where that landmark was located (Fig 4A, middle), akin to 'landmark' cells in the hippocampus [41]. The majority (65.8%) of the units had 'conjunctive' response patterns that were modulated by both space and landmark position, which are reminiscent of hippocampal cells that encode conjunctions between location and sensory observations [44,45] (Fig 4A, bottom). The patterns of response to space and objects in the recurrent units of the RL agent were thus similar to those observed in neurophysiological recordings made whilst rodents navigate.

An alternative method for classifying activation patterns is by examining their stability across time points within and across environments [51]. We observe similar results when classifying cells in this way: 'vector' units are more likely to have spatial patterns that are stable across environments, while 'transition' units are more likely to have spatial patterns that remap across different environments (S9 Fig). The stability of spatial representations in 'vector' units hint at a role for these units in structural generalization across spatial environments [26]. Notably, despite the proposed role of grid cells in vector-based navigation [24,25], we do not observe strong evidence for grid cells in our networks (see S10 Fig for analysis and discussion).

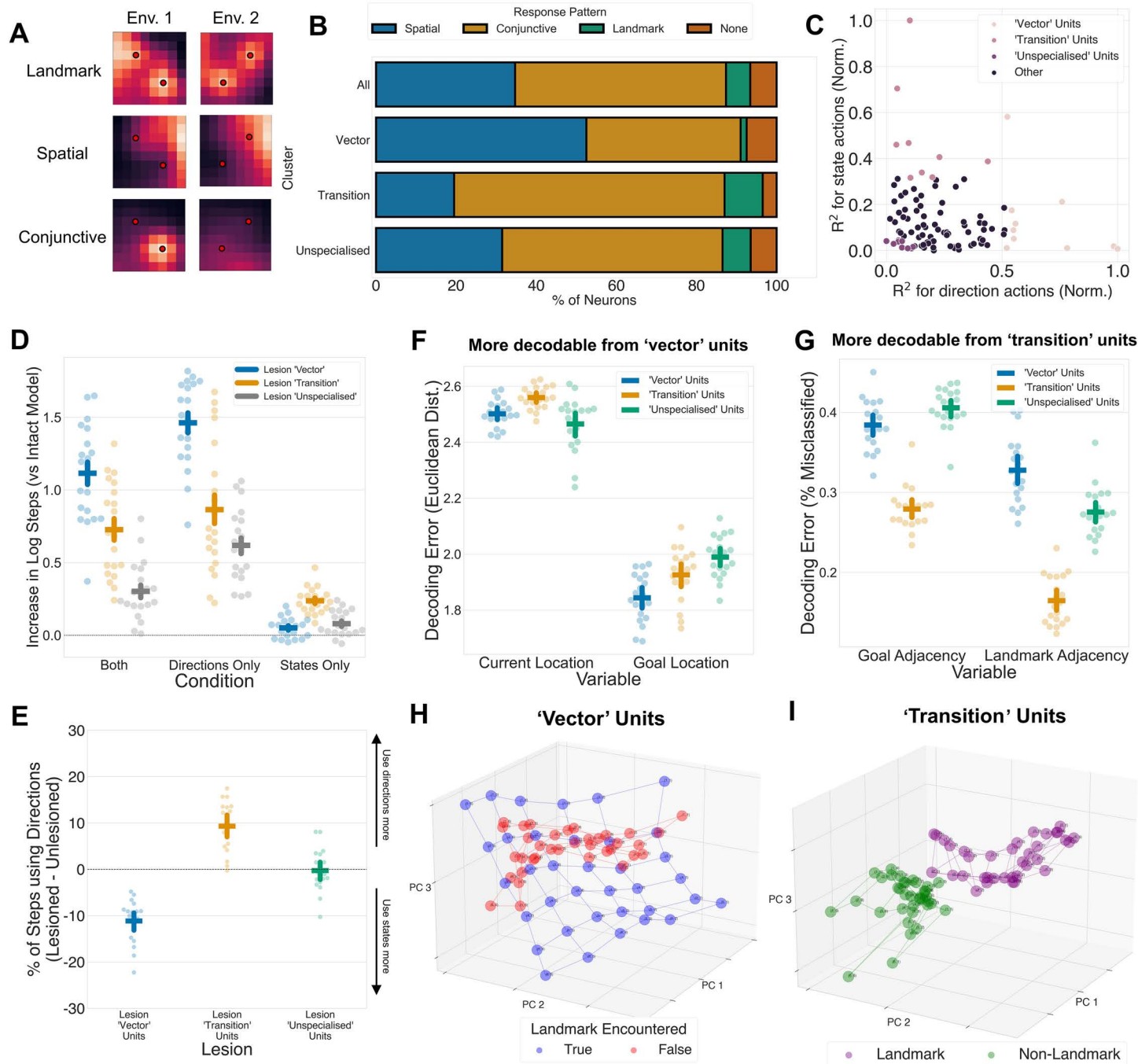

**Fig 4. Models spontaneously develop separate modules for 'vector-' and 'transition'-based strategies. A:** Example heatmaps for a unit with a 'landmark', 'spatial', or 'conjunctive' response pattern in two different environments. Red circles on the heat maps denotes the presence of a landmark in that location. 'Spatial' units respond stably to certain regions of space across environments regardless of landmark configuration, 'landmark' units respond to all landmarks across environments, while 'conjunctive' units respond to landmarks differently across different environments. **B:** Stacked bar plots showing the proportion of response pattern types across all units or in the 'vector', 'transition', or 'unspecialized' clusters. 'Vector' units are more likely to have spatial responses, while 'transition' units are more likely to have conjunctive or landmark responses. **C:** Scatter plot showing the $R^2$ value for the correlation between the activation of each LSTM unit's cell state and the output values of the either the 'direction' or 'state' actions in the policy network. Values are normalized to be between 0 and 1. The 20 units that explained the most variance in 'direction' or 'state' actions were designated as the 'vector' and 'transition' units, respectively, while the 20 units that explained the least variance in either type of action were designated as

'unspecialized' units. **D:** Performance deficit of lesioned models (as measured by excess number of steps taken to get to the goal compared to an intact model) on the *both, directions-only*, and *states-only* conditions. Each dot represents one of the 20 trained models, and the line represents the mean, and the error bar represents the 95% confidence interval. Lesioning 'vector' units leads to deficits in the *'directions-only'* condition, while lesioning 'transition' units leads to deficits in the *'states-only'* condition. Lesioning 'vector' and 'transition' units both lead to deficits in the *'both'* condition. **E:** Change in use of 'direction' actions after lesions to the 'vector', 'transition' and 'unspecialized' units, compared to the unlesioned models. Each dot represents one of the 20 trained models, and the line represents the mean, and the error bar represents the 95% confidence interval. Lesioning 'vector' units leads to a decrease in use of 'direction' actions and lesioning 'transition' units leads to a decrease in use of 'state' actions. **F:** Decoding error on held-out time steps for current and goal locations (as measured by Euclidean distance) for the 'vector', 'transition' and 'unspecialized' units. Current and goal locations are both best decodable from 'vector' units. **G:** Decoding error on held-out time steps for whether the agent is currently adjacent to a landmark or a goal. Goal and landmark adjacency are both best decodable in 'transition' units. **H:** First three principal components for the PCA on the cell state activations of 'vector' units. Each dot represents the centroid of the PCs for each location in the grid. Red and blue dots represent the PCs before and after a landmark is encountered, respectively. The representations of 'vector' units faithfully reflect spatial structure after a landmark is encountered. PCA results are shown for one representative model. **I:** First three principal components for the PCA on the cell state activations of 'transition' units. Each dot represents the centroid of the PCs for each location in the grid. Purple and green dots represent the PCs for non-landmarks and landmarks, respectively. The representations of 'transition' units seem to separate landmarks and non-landmarks without apparent spatial structure. Data and code underlying this figure are available at https://osf.io/w39d5/ and https://github.com/denis-lan/navigation-strategies, respectively.

Next, we asked whether these response patterns in the LSTM causally influenced the ability to use vector- and transition-based navigational strategies. To test this, we regressed the output logits of either the 'directions' or the 'states' actions in the policy head on the cell state activations of all LSTM units. We identified the 10 units whose cell state responses explained the most variance in either the 'directions' or 'states' outputs as either 'vector' units or 'transition' units, respectively (Fig 4C). As a control, we also selected the 10 units that explained the least variance in both 'vector' or 'transition' outputs as 'unspecialized' units.

Lesioning the 'vector', 'transition' and 'unspecialized' units had different consequences for performance on the various conditions of the task (Fig 4D; *ANOVA*: lesion × condition: $F(4, 76) = 34.09$, $p < .0001$). Crucially, lesioning the 'vector' and 'transition' units led to a double dissociation in the performance of RL agents in the '*directions only*' and '*states only*' conditions. Lesioning 'vector' units led to worse performance on the '*directions only*' condition than lesioning 'transition' (Bonferroni-corrected *t* test: $t(19) = 5.12$, $p < .0001$) and 'unspecialized' units ($t(19) = 10.31$, $p < .0001$). By contrast, lesioning 'transition' units led to worse performance on the *states only* condition than lesioning 'vector' ($t(19) = -9.27$, $p < .0001$) and 'unspecialized' units ($t(19) = 7.27$, $p < .0001$). Moreover, lesioning both 'vector' and 'transition' units lead to a greater performance deficit compared to lesioning 'unspecialized' units when agents could freely choose between either strategy ('vector' versus 'unspecialized': $t(19) = 12.76$, $p < .001$; 'transition' versus 'unspecialized': $t(19) = 5.23$, $p = .002$). In this condition, these deficits likely resulted from different perturbations in agents' strategy use (Fig 4E; $F(3, 57) = 88.07$, $p < .0001$). Lesioning 'vector' units pushed the agents away from using 'direction' responses ($t(19) = -7.61$, $p < .0001$), while lesioning 'transition' units pushed agents away from using 'state' responses ($t(19) = 6.44$, $p < .0001$). Overall, these results suggest that 'vector' and 'transition' units were indeed causally responsible for implementing vector-based and transition-based strategies, respectively.

We predicted that 'vector' and 'transition' units would exhibit different patterns of response to spatial locations and landmarks. For example, 'vector' units need to know about the spatial structure across environments and so might be more prone to exhibit stable spatial responses, whereas 'transition' units that exploit environment-specific landmark information might additionally be modulated by the presence of landmarks. Indeed, a chi-squared test revealed a significant association between the type of unit (i.e., 'vector', 'transition', or 'unspecialized') and response pattern (i.e., 'spatial', 'landmark', or 'conjunctive'; $\chi^2 = 62.26$, $p < .001$, $d.f. = 6$, $n = 600$). 'Vector' units were significantly more likely to have 'spatial' response patterns (adjusted Pearson residual = 4.33, $p = .0002$) and less likely to have 'conjunctive' (adjusted Pearson residual = −2.93, $p = .003$) or 'landmark' (adjusted Pearson residual = −2.60, $p = .009$) response patterns, while 'transition' units were significantly less likely to have 'spatial' response patterns (adjusted Pearson residual = −3.61, $p = .0003$) and more likely to have 'conjunctive' (adjusted Pearson residual = 2.67, $p = .007$) or 'landmark' (adjusted Pearson residual = 2.02,

*p* = .04) response patterns (Fig 4B). In other words, 'vector' units are less prone to remap, whereas 'transition' and 'unspecialized' units show responses that are linked to salient objects, and that remap across environments [52]. Overall, this pattern of results is reminiscent of models and data suggesting that the prefrontal and entorhinal cortices maintain stable structural representations across environments and tasks, while hippocampal representations are influenced by environmental or task specifics [26,53].

The different activation patterns of these units likely reflect the fact that 'vector-'and 'transition'-based strategies involve computations involving different task variables. To determine which variables were encoded by 'vector' and 'transition' units, we attempted to decode a range of task variables from the cells state activations of each class of unit, evaluating accuracy on held out data. These analyses revealed that 'vector' units had lower decoding errors for both the current location (*ANOVA: F*(2,57) = 10.64, *p* < .0001, *post-hoc t test for 'vector' versus 'transitions': t*(38) = −4.25, *p* = .0003) and the goal location (*F*(2,57) = 15.24, *p* < .0001; *t*(38) = −2.95, *p* = .01; Fig 4F). However, 'transition' units had lower decoding error for landmark adjacency (i.e., whether the agent was next to a landmark; *F*(2,57) = 125.21, *p* < .0001; *t*(38) = −14.16, *p* < .0001) and goal adjacency (*F*(2,57) = 125.21, *p* < .0001; *t*(38) = −11.72, *p* < .0001; Fig 4G).

These analyses strongly imply that 'vector' and 'transition' units represent the environment differentially. To visualize their representations, we performed a principal component analysis (PCA) on the cell state responses of the 'vector' and 'transition' units. The PCA on the 'vector' units suggested that the representational geometry of 'vector' units seemed to respect spatial structure, especially after a landmark had been encountered (Fig 4H). This re-organization in representational geometry to reflect spatial structure after encountering a landmark is reminiscent of models that suggest that 'landmarks' bind knowledge of the graph structures with knowledge of abstract spatial structures [8, 43]. On the other hand, 'transition' units seemed to separate landmarks and non-landmarks as they were encountered, but without any apparent spatial structure (Fig 4I). Interestingly, 'unspecialized' units seemed to do a combination of both, representing location in 'landmarks' and 'non-landmarks' separately on orthogonal maps (S11F Fig)

## Experiment 3: Human participants successfully sample maps for few-shot navigation

In Experiments 1 and 2, participants were obliged to learn about landmarks by viewing objects within the map at randomly pre-selected locations. However, in many real-life few-shot planning scenarios, humans can freely choose which states to use as landmarks, such as by choosing which landmarks to attend to when reading a map. For example, when navigating a new city, one might choose to learn the location of a few well-placed landmarks, which might serve navigation better than randomly placed landmarks. Thus, in Experiment 3, we asked whether participants would perform even better when they could decide where to sample the maps themselves in the map-reading phase. Two groups of participants participated in Experiment 3, which was held across two days. On the first day, participants both performed a task identical to Experiment 2, except in an open-field environment. On the second day, we varied the sampling conditions during the map reading phase. One group (*n* = 100) of participants was allowed to freely sample the map, with a budget of 16 samples (re-sampling of locations allowed) in the learning phase (*free sampling* group). The other group (*n* = 101) also viewed 16 samples, but these were (randomly) pre-selected for them, as in previous experiments (*forced sampling* group). Once again, we replicated the key behavioral results from previous experiments, including the use of state responses at landmarks and goals and the quadratic relationship between direction use and performance (S3 Fig)

On day 2, participants in the *free sampling* group performed better than those in the *forced sampling* group (Fig 5A). A mixed effects linear model predicting log-transformed steps to goal on the second day of the study showed that this result held even after controlling for performance on the first day of the study (*β* = −0.095, *SE* = 0.025, *t*(172.93) = −3.76, *p* = .0002). In other words, people can adaptively select locations that are likely to help them navigate in the subsequent task.

Where did this benefit of free sampling come from? To address this question, we devised three 'landmark sampling' metrics and investigated their relationship with navigation performance. These were (1) the average distance from each

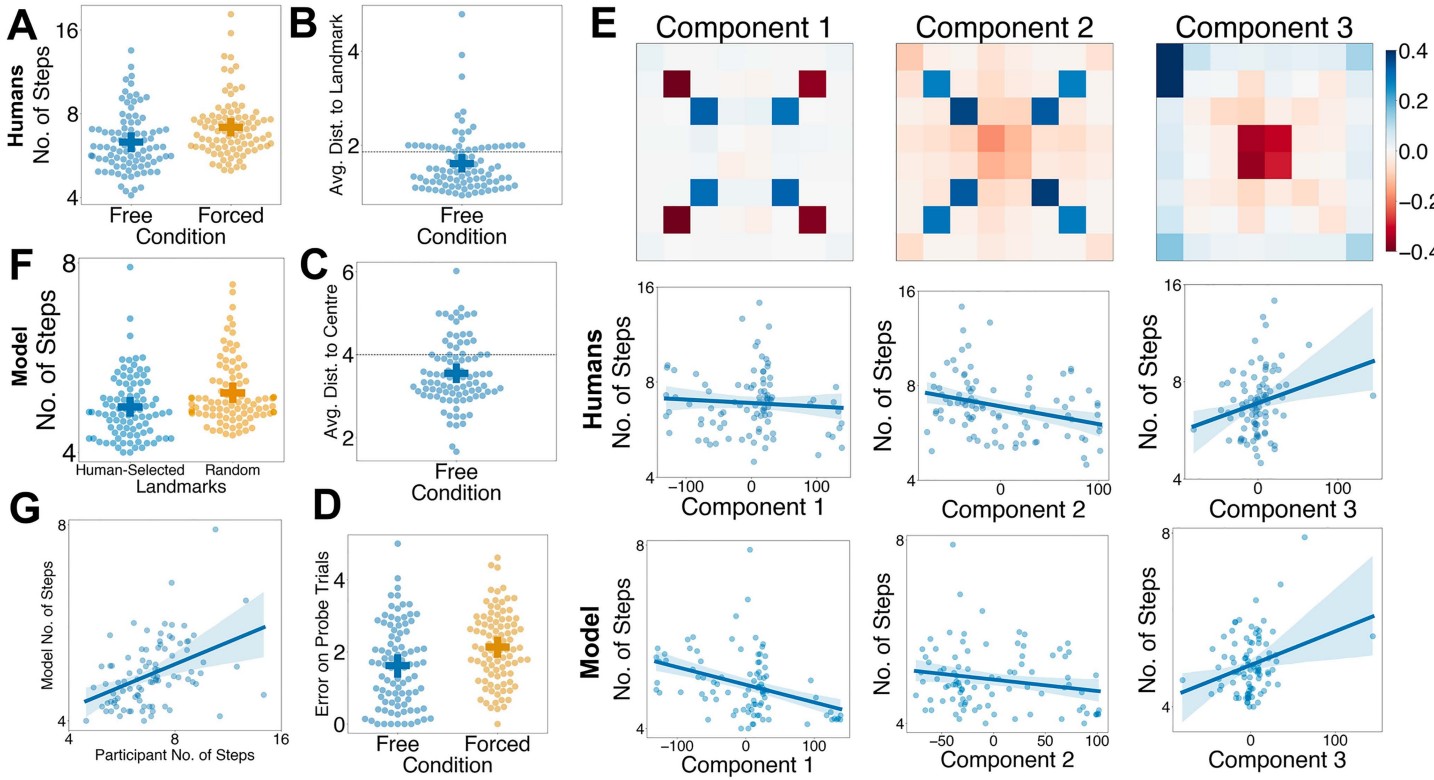

**Fig 5. Participants successfully choose landmarks that are beneficial for few-shot navigation. A:** Performance on the second day in the free-sampling and forced-sampling groups as measured by number of steps taken to get to a goal (y-axis, represented on a logarithmic scale). Each dot represents an individual participants' mean performance across the task and the dash represents mean performance across all participants, with the error bars representing the 95% CI. **B:** Average distance from all states to their nearest landmarks for participants in the free-sampling group. The dotted line represents the mean distance expected by chance. **C:** Average distance from landmarks to the center for participants in the free-sampling group. The dotted line represents the mean distance expected by chance. **D:** Mean error on probe trials for participants in the free-sampling and forced-sampling conditions. **E:** Results from a principal component analysis (PCA) conducted on each participants' number of samples on each location in the 8 × 8 grid. The loadings for each of the locations in the 8 × 8 grid on the three sampling-related principal components are shown in the first row. The second row shows the relationship between each component (x-axes) and navigation performance, as measured by number of steps taken to reach a goal (y-axes, represented on a logarithmic scale). Each dot represents an individual participant, and the line represents the best-fitting line. The third row shows the same for the deep RL model, with each dot represents the model's performance when tested on the landmarks sampled by an individual human participant. **F:** Model performance, as measured by steps taken to goal (y-axis, represented on a logarithmic scale) when tested on the freely selected landmarks chosen by the free-sampling group or the randomly chosen landmarks that the forced-sampling group was exposed to. Each dot represents the model's performance when tested on the landmarks a single human participant sampled or was exposed to. **G:** Correlation between participant's performance (as measured by number of steps taken to reach the goal, represented on a logarithmic scale) and the model's performance when it is tested on the landmarks chosen by each participant. Participants' performance on the navigation phase was significantly associated with the models' performance when it was tested on the landmarks they chose. Data and code underlying this figure are available at https://osf.io/w39d5/ and https://github.com/denis-lan/navigation-strategies, respectively.

sampled landmark to every other state, (2) the average distance from each sampled landmark to the center, and (3) the average error on a series of probe trials, conducted at the end of a few navigation trials, where participants were asked to recall the location of sampled landmarks. We expected that the superior performance in the free-sampling condition might occur because landmarks were sampled to be well-distributed and thus accessible from all states in the environment, central, and easier to remember.

Indeed, participants did choose landmarks such that the distance from all states to the nearest landmark was lower than would be expected under our random landmark manipulation in the forced-sampling group ($M = 1.61$, $t(91) = -3.81$,

*p* = .0002; Fig 5B). Moreover, they chose landmarks that were closer to the center of the grid than would be expected (chance = 4, *M* = 3.55, *t*(91) = −5.47, *p* < .0001; Fig 5C). Participants in the free sampling condition performed better on probe trials than participants in the forced sampling condition, even after controlling for probe trial performance on the first day (linear mixed effects model: *β* = −0.32, *SE* = 0.12, *t*(163.3) = −2.62, *p* = .009; Fig 5D). Across all participants in the free-sampling condition, we found that after controlling for performance on the first day, participants' navigation performance on the second day of the task as measured by log-transformed steps to goal was significantly predicted by all three measures. Participants tended to reach the goal in fewer steps when they arranged landmarks in a way that reduced the average distance from all states to their nearest landmarks (*β* = 0.075, *SE* = 0.031, *t*(87) = 2.44, *p* = .017), chose landmarks that were closer to the center (*β* = 0.069, *SE* = 0.022, *t*(87) = 3.07, *p* = .003), and had smaller errors on landmark location probe trials (*β* = 0.10, *SE* = 0.017, *t*(87) = 6.13, *p* = .003). Overall, participants sampled maps such that landmarks were more accessible, central, and memorable, all of which facilitated later navigation.

Using a complementary, data-driven approach, we also performed a principal component analysis (PCA) on participants' free sampling locations. The loadings reflected interpretable aspects of participants' sampling strategies (Fig 5E). The first principal component (PC) reflected a distinction between two dominant strategies for exploration, each focusing on either the four states diagonally adjacent to the four central states or the four states diagonally adjacent to the four corner states. The second PC reflected a distinction between strategies that focused on these eight squares or any other state in the grid (especially the central and corner states). The third component reflected a distinction between strategies that focused on central states compared to those that focused on states at the corners and edges.

Subsequently, we asked whether between-participant variation in PC loadings was associated with better performance on the navigation phase. We performed a linear regression in which log-transformed steps to goal (on the second day) was predicted the three sampling-related PCs, with the mean log-transformed steps to goal on the first day of the task as a nuisance variable. Both the second sampling-related component (*β* = −0.046, *SE* = 0.020, *t*(87) = −2.33, *p* = .02) and the third sampling-related component (*β* = 0.073, *SE* = 0.019, *t*(87) = 3.76, *p* < .001) significantly predicted performance on the navigation phase. While the third PC seems obviously linked to the centrality of landmark states, the second PC might reflect participants attempting to minimize the average distance to the closest landmark (S13B Fig). These two PCs were correlated with the hypotheses-driven metrics, such as the accessibility, centrality, and the memorability of landmarks (S13C Fig).

If humans' landmark choices were indeed beneficial for navigation, we would expect that our deep reinforcement learning model would also perform better when navigating with human-selected landmarks than with random landmarks. Indeed, we found that our deep RL model did indeed reach the goal in fewer steps when navigating with human-selected landmarks (mixed effects model: *β* = −0.051, *SE* = 0.017, *t*(183) = −2.96, *p* = .003; Fig 5F). Within the 'free' choice condition, there was a strong correlation between each participant's performance on the task and the model's performance when tested on the landmarks chosen by the respective participants (linear regression: *β* = 0.22, *SE* = 0.05, *t*(90) = 4.47, *p* < .001; Fig 5G). Moreover, the exploration-related PCs also predicted model performance in similar ways (Fig 5E)—like in human participants, the second PC (linear regression: *β* = −0.022, *SE* = 0.011, *t*(88) = −2.09, *p* = .04) and third PC (*β* = −0.034, *SE* = 0.011, *t*(88) = 31.18, *p* = .002) also predicted navigation performance in models. Additionally, the first PC (*β* = −0.043, *SE* = 0.011, *t*(88) = −4.04, *p* < .001) also predicted navigation performance in the models, although it did not significantly predict navigation performance in humans.

## Discussion

Across three behavioral experiments, we show that humans use a mixture of vector- and transition-based strategies during few-shot navigation. Specifically, people rely predominantly on vector-based strategies to head in the general goal direction, but fine-tune their navigation with transition-based strategies when near landmarks or the goal. This is consistent with theories proposing that both topological and geometrical cognitive maps are both encoded in neural states [8], and suggests that humans can flexibly switch between representations depending on environmental demands. The crucial

role of 'landmarks' in our task is in line with previous work suggesting an important role for landmarks in localization [7,54,55], particularly during path integration or 'dead reckoning' [56,57]. In other words, 'landmarks' might help anchor graph-like representations of objects in the environment to map-like, Euclidean representations of space [8]. Furthermore, we show that human participants perform better when they freely select their own landmarks. This is likely due to their ability to select well-positioned landmarks, although it is also possible that the act of active sampling itself might also lead to an advantage in spatial memory [58,59].

Meta-learning has recently emerged as a popular framework for the rational analysis of cognition [35,60]. Our work applies a meta-learning framework to study human navigation. We show that deep meta-reinforcement learning models trained on our few-shot navigation task develop behavioral policies that resemble those of humans. This implies that humans and networks develop rational inductive biases for rapidly solving navigational problems on unfamiliar maps. In humans and other animals, these biases are likely shaped by both evolution and development. Indeed, early-life navigational experience, such as the structure of street networks in the place one grew up, has been shown to exert a strong influence on individuals' navigational ability and strategy use [61,62]. We used the same environments for training all our meta-learning agents, and so they developed relatively homogenous strategies. However, an interesting extension of our work might be to investigate whether the types of environments that meta-learning agents are trained on could account for the diversity in human navigational ability and strategy use we observe in our task and other navigational tasks. For example, while meta-learning facilitates structural generalization in similar environments, it would be interesting to see when and how generalization breaks down in environments that lack the underlying structure seen during training.

Aside from the similarities to human behavior, we find that the meta-learning networks naturally develop distinct ways of representing environments that are important for implementing vector- and transition-based strategies. These representations are reminiscent of the diverse representations and neural response patterns that have been previously observed in mammalian navigation systems [8,26,39,44], implying that these representations in mammalian brains might also have developed for implementing different strategies during flexible navigation. These results open up the possibility for further neural investigations that delve deeper into how the representations developed by these networks might map onto the different representations that might exist in human brains during navigation. Of course, while we have relied solely on reward-maximization objective to train our networks, it is unlikely that reinforcement is sufficient to account for the diversity of neural representations observed in mammals during navigation. Previous work has found the supervised [25] and self-supervised or predictive [26,63,64] objectives lead to the emergence of stronger brain-like representations, and future work should investigate how a combination of different objectives affects networks' use and implementation of different types of navigational strategies.

Our experiments involved a deliberately stylized environment involving navigation on an abstract, spatially organized grid, which allowed us to cleanly dissociate between the use of vector- and transition-based strategies. However, this meant that our task lacks many aspects of naturalistic spatial navigation, such as the need to transform egocentric sensory inputs to allocentric frames of reference. At the same time, participants in our task could only observe states that they were adjacent to, while real-life navigation often involves using distal states as reference points [65–67] or, when the goal is visible, targets to travel towards (i.e., 'beaconing') [68]. Importantly, while we artificially dissociate between vector and state transition-based strategies, these strategies are likely not so distinct in real-world navigation, where we likely integrate information across both spatial and associative structures. Overall, while we expect that naturalistic navigation will rely on many of the same principles, future experiments using more naturalistic paradigms, such as virtual reality [69], should be conducted to confirm whether our findings extend to real-life navigational settings.

While we have focused on planning in spatial settings, the principles we illuminate likely generalize to planning more generally as well, especially given the apparent overlap between neural representations of spatial and non-spatial concepts [28,70]. Indeed, the distinction between environment-specific associative knowledge in the hippocampus and generalizable spatial representations in the entorhinal cortex and other cortical areas is reminiscent of frameworks in

psychology and neuroscience positing the existence of distinct learning or 'memory' systems [71]: a fast-learning store for context-specific information in the medial temporal lobe, and a gradual learning store for generalizable structure in the neocortex. Flexible planning in non-spatial domains likely requires us to exploit both types of knowledge, perhaps using similar principles: exploiting transition structure near familiar, 'landmark' regions of state space, but generalizable structure (e.g., heuristics) whenever knowledge of the transition structure is lacking.

## Materials and methods

Human behavioral experiments were approved by the University of Oxford Central University Research Ethics Committee (reference number: R50750/RE001). All participants provided consent via an online consent form. These experiments were conducted according to the principles expressed in the Declaration of Helsinki.

### Experiment 1

**Participants.** Participants were recruited through the online platform Prolific. The inclusion criteria for being invited to participate were as follows: reported age between 18 and 40 (inclusive), English as first language, approval rate on Prolific more than 85%, and did not participate in a previous version of the study. Participants were offered £9 for completing the task, with the opportunity of earning up to £3 of bonus for their performance on the task. 200 participants participated in the experiment in total, with 100 participants each participating in the open field and cluttered versions of the experiment. Participants who performed more than 2 standard deviations below the mean performance were then excluded, where performance was measured by log-transformed number of steps taken to reach the goal. Two participants were hence excluded from the open field condition and two participants were excluded from the cluttered condition.

### Procedure

**Pre-task questionnaires and instructions.** Participants provided informed consent through an online consent form. Before completing the task, all participants completed the Santa Barbara Sense of Direction Scale [72] and the Navigation Strategy Questionnaire [73]. Participants read the instructions before the start of the experiment and were asked to do a practice run of each phase to ensure they fully understood the instructions. Participants were instructed that they should not use memorization aids (e.g., writing things down) to complete the task. If participants needed a refresher of the instructions, participants could press a button to re-read the instructions at any time during the practice phase and the main task.

**Task set-up.** Participants completed a study in which they navigated through an 8 × 8 grid of pictures of objects (Fig 1A). These images were selected from the Bank of Standardized Stimuli [74]. Every trial involved a grid with completely different sets of images in different locations. Each grid square contained either an object or an obstacle. Every object was represented by a unique image, while every obstacle was represented by an image of a boulder.

There were two *environment* conditions that determined how obstacles were placed. In both conditions, the outer squares of the grid contained obstacles, forming boundary walls. Thus, although the entire environment was a 10 × 10 grid, the traversable interior of the grid was 8 × 8. In the 'open field' condition, there were no further obstacles. In the 'cluttered' condition, there were obstacles placed at pseudorandom positions within the confines of the walls (generated such that there is one obstacle in each row and each column, i.e., 8 boulders in total). The locations of these interior obstacles were regenerated on every trial.

Participants completed 4 blocks, each consisting of 8 trials. Every trial of the task involved two phases: a 'map reading' phase and a 'navigation' phase.

**Map reading phase.** In this phase, participants learnt about the location of a limited (2, 4, 8, or 16) number of landmarks and the location of the goal object. 'Landmark' and 'goal' locations were chosen at random at the start of each trial. The only constraint to their selection was that all locations were unique grid squares that were not on the edge of the 8 × 8 traversable grid. Within each block of trials, the total number of trials with 2, 4, 8 or 16 landmarks was balanced (2 of each per block) and occurred in random order.

Participants learnt about 'landmark' and 'goal' locations by interacting with a 'map' of the environment. This 'map' was represented as a birds' eye view of the grid. Specifically, participants viewed a 10 × 10 grid (the 8 × 8 interior of the grid and the surrounding walls) displayed centrally on the screen. Grid squares containing objects were blanked out so that its object was unidentifiable. Squares containing boulders were presented clearly so that obstacle locations were known to the participant.

Participants first learnt about the 'landmark' objects and their locations. A subset of the grid squares was successively highlighted in blue (S1A Fig). Participants were required to click on these grid squares in the sequence that they appeared. Each grid square clicked immediately revealed a 'landmark' object at the corresponding location. Clicks elsewhere had no effect. Every successive blue square appeared 1 second after participants clicked on the previous blue square and remained on the screen until participants had clicked on it. The revealed object remained on the screen until participants clicked on the next square, or for 3 s if participants had not clicked on the next square after 3 s.

Trials could comprise a variable number of unique blue square locations (2, 4, 8, or 16), but the number of total presentations were kept the same (2 landmarks would be shown 8 times each, 16 landmarks would be shown once each, etc.). The order of the clicks was pseudorandom with two constraints: (1) no landmark was shown twice consecutively, and (2) each unique landmark was presented an equal number of times before any landmark could be repeated. A counter located at the bottom of the screen ("Clicks Left:") indicated the number of clicks (on blue squares) that remained.

After learning about the 'landmark' objects, participants learnt about the 'goal' object. 1 second after participants clicked on the final blue square, a single grid square appeared highlighted in yellow. Participants were required to click on the yellow grid square to reveal the 'goal' object for the upcoming trial. Text instructions at the top of the screen reminded participants that the yellow grid square was the 'goal' for the upcoming navigation phase. 1 second after participants clicked on the yellow grid square, a 'Done' button would appear allowing participants to terminate the map-reading phase. The revealed goal object remained on the screen until participants clicked on the 'Done' button, or for 3 seconds if participants had not clicked on the 'Done' button after 3 s.

In the practice run of the map reading phase, participants were tasked with clicking on three successive blue grid squares to reveal the landmark objects, followed by a yellow grid square to reveal the goal object.

**Navigation phase.** In the 'navigation' phase of each trial, participants were required to navigate to the goal object from a random start location (S1C Fig). The goal state was the object revealed by clicking on the yellow square on the preceding map reading phase. The start location was chosen to be a random location on the grid that 4 steps away from the goal. The start location was chosen not to be a landmark object, goal object, or obstacle, and could not be on the edge of the grid. Participants began the trial with 1,000 points. Every step participants took cost 50 points, and reaching the goal state earned participants 1,000 points. The total amount of points determined the amount of bonus payment participants would receive at the end of the task, so participants were incentivized to reach the goal as soon as possible.

Participants viewed objects denoting their current state (in a central display) and the goal state (shown at the top on the screen). The central display ('viewfinder') had the current state in a central square was surrounded by four squares on each edge, with red arrows indicating the four possible directions. These four red arrows were always present in the viewfinder, even when there were walls or obstacles in one of the four directions.

**Navigation methods.** Participants could navigate in two ways via displays located on the left or right of the 'viewfinder'. To the right (or left; counterbalanced over blocks) were four arrows pointing up, down, left and right, each displayed in a box in the corresponding position with respect to a central box ('*directions* display'). To the left of the screen, the objects corresponding to the four states that were cardinally adjacent to the current state (up, down, left right; this could include obstacles as images of boulders if they were adjacent to the current state) were shown ('*states* display'). Each object was shown in a box in a 2 × 2 grid, and the location of each object was randomized such that their position was uninformative of their location with respect to the current object.

Participants could navigate to the states adjacent to the current state by either clicking on an arrow from the *directions* display or an object from the states display. Clicking on one of the arrows moved participants one step in the direction of

the arrow. Alternatively, clicking on one of the adjacent objects on the '*states* display' brought participants to the state corresponding to the clicked object. Clicks which moved participants in the direction of an obstacle had no effect and incurred no cost in points. Clicking on an arrow or an object were equivalent in that they only allowed movement to the same adjacent states. However, having these two different movement methods allowed us to determine whether participants were focusing on the direction in which they were going (using a 'vector-based' strategy), or the identity of the next state that they would be transitioning to (using a 'transition-based' strategy).

After clicking on an arrow or an object, participants viewed an animation in the viewfinder showing the direction in which they had travelled in and the next object they traversed to. This animation proceeded as follows: all the red arrows in the 'viewfinder' except the one corresponding to the movement direction disappeared, the next object appeared in the box corresponding to the movement direction, and the current object was replaced by the next object. This animation lasted for 3 s, after which the 'viewfinder' returned to its default state with the new object in the central square, and the images in the '*states* display' were updated to show the objects adjacent to the new state. The animation was identical regardless of whether participants chose an arrow or an object. In other words, participants received identical information about the movement direction and the new state regardless of whether they clicked on an arrow or an object.

**End of the navigation phase.** The navigation trial terminated only when participants had reached the goal state. When making a step to the goal state, participants saw the animation (as described above) for reaching the goal state. After the animation ended, a bell sound effect played indicating participants had reached the goal state, and a green text reading "+1000" appeared indicating participants had earned 1,000 points. The screen then progressed automatically to either the map reading phase of the next trial, or the instructions screen for the start of the next block if this were the last trial in the block. The experiment terminated at this point if this were the last trial of the last block.

Before performing the main task, participants were allowed a practice run. In the practice run of the navigation phase, participants were given the opportunity to interact with both the '*directions* display' and the '*states* display'. In the practice run, there was no goal to navigate to, and participants were allowed to experiment with the different movement methods by taking 10 steps around the grid. On the first step, the '*directions* display' was disabled and participants were forced to make a step using the '*states* display'. On the second step, the '*states* display' was disabled and participants were forced to make a step using the '*directions* display'. On subsequent steps, none of the displays were disabled, and participants could make a step using any movement method they wanted. The practice navigation phase terminated once the participant had made 10 steps around the grid.

**Navigation phase conditions.** Participants performed four task conditions, one in each block, in random order, using a within-participants design. Each condition varied the navigation methods that were available. Participants were instructed as to which condition they were about to experience in the instructions screen at the start of each block.

In the '*both*' condition, all navigation methods were always available and the navigation proceeded exactly as described above, with both '*directions*' and '*states*' displays being available for participants to click on throughout the whole trial.

In the '*directions only*' condition, participants could only click on the '*directions* display'. The '*states* display' was greyed out to indicate that participants could not click on the objects, but the images in the object display continued to update throughout the task (S1D Fig). Text instructions on top of the '*states* display' indicated that clicking on the objects was currently disabled.

In the '*states only*' condition, participants could only click on the '*states* display'. The '*directions* display' was greyed out to indicate that participants could not click on the arrows (S1E Fig). Text instructions on top of the '*directions* display' indicated that clicking on the arrows was currently disabled.

In the '*random alternation*' condition, participants were restricted to either *directions* or *states* on each step, with the type of response available being randomly chosen on each step. When a movement method was unavailable, the corresponding display was greyed out to indicate that participants could not interact with it. Text instructions above the disabled display also indicated that the display was currently disabled. The percentage chance for each type of response was set

based on the proportion of *direction* responses made when participants could freely choose between response strategies in pilot experiments: this was 75% chance of *direction* responses being available in the open field version, and 50% chance in the cluttered version. Note that in the full dataset, participants in the cluttered condition ended up using *direction* responses at a rate of 63.6%.

**Debriefing.** After completing the task, participants were asked to describe what strategies they used to complete the task in the different task conditions and whether they had used memorization aids (e.g., writing things down) to complete the task. Participants were told that admitting to using memorization aids would not affect their compensation for completing the task. No participants were excluded for admitting to using memorization aids on Experiment 1.

## Experiment 2

**Participants.** We recruited 100 participants using the same recruitment methods as Experiment 1. Participants were excluded if they experienced technical difficulties that hampered their performance on the task or indicated in post-task questionnaires that they had used external memorization aids (e.g., writing things down) to complete the task, and new participants were recruited to make up the numbers. We excluded from analysis participants who performed more than 2 standard deviations worse than the mean performance for all participants in each condition, as measured by log-transformed number of steps taken to reach the goal. 4 participants were excluded on the basis of performance. The recruitment method, exclusion criterion, task structure, and some analyses were pre-registered on 11 December 2023 (pre-registration document available at https://osf.io/eu89x [38]).

### Task procedure

**Pre-task questionnaires.** As in Experiment 1, participants provided informed consent through an online consent form. Before completing the task, all participants completed the Santa Barbara Sense of Direction Scale and the Navigation Strategy Questionnaire.

**Map reading and navigation phase.** The map reading and navigation phases in Experiment 2 proceeded in an identical way to Experiment 1. The differences in Experiments 1 and 2 concerned the environmental and action conditions, as well as the number of landmarks. All participants in Experiment 2 performed the task in a cluttered environment. Both *direction* and *state* responses were available throughout the whole experiment (i.e., analogous to the 'Both' condition in Experiment 1).

Participants performed 4 blocks of 6 trials each. Unlike Experiment 1 where the number of landmarks was interleaved, the number of landmarks was blocked in Experiment 2. Each block had a different number of landmarks: either 2, 4, 8, or 16 landmarks in shuffled order. The methods for selecting and presenting landmarks were identical to Experiment 1. All other aspects of the experiment, including the structure of the map reading and navigation phases, proceeded identically to Experiment 1.

The practice run for the map reading phase proceeded in an identical fashion as Experiment 1. The practice run for the navigation phase proceeded in a similar fashion as in Experiment 1, where participants did not have a goal and were allowed to experiment with the navigation methods by moving 10 steps around the grid. However, as this experiment only involved the 'Both' condition, both the *'directions* display' and the '*states* display' was enabled throughout the practice run. Nevertheless, participants were asked to experiment with both displays to familiarize themselves with both navigation methods. If participants had not used one of the two displays at the end of 10 steps, the practice trial would continue until participants had tried using that display.

**Debriefing.** After completing the task, participants were asked to describe what strategies they used to complete the task and whether they approached the task differently when there were different numbers of landmarks. They were also asked whether they had used memorization aids (e.g., writing things down) to complete the task. Participants were told that admitting to using memorization aids would not affect their compensation for completing the task. Two participants were excluded for admitting to using memorization aids, and new participants were recruited so that there were 100

participants in total. This protocol for excluding and recruiting new participants was pre-registered in our pre-registration document [38].

## Experiment 3

**Participants.** We recruited participants using the same recruitment methods as Experiments 1 and 2. Experiment 3 occurred over two consecutive days. There were two conditions in Experiment 3: a 'free-sampling' condition and a 'forced-sampling' condition. 100 participants in the free-sampling condition and 101 participants in the forced-sampling condition completed both days of the experiment. Participants were excluded from analysis if they performed more than 2 standard deviations worse than the mean performance (averaged over both days of the task) for all participants in the same condition, as measured by log-transformed number of steps taken to reach the goal. 8 participants ended up being excluded from analyses in each condition, leaving data from 92 participants in the free-sampling group and 93 participants in the forced-sampling group.

## Task procedure

### Day 1

**Pre-task questionnaires.** As in Experiments 1 and 2, participants provided informed consent through an online consent form. Before completing the task, all participants completed the Santa Barbara Sense of Direction Scale and the Navigation Strategy Questionnaire. They also read the instructions and completed practice runs of the map reading, navigation, and memory probe phases.

**Map reading and navigation phases.** All participants completed the same task on Day 1 of the study. All participants completed a task similar to Experiments 1 and 2. The map-reading and navigation phases proceeded identically to Experiments 1 and 2, but there were occasional additional memory probe phases (detailed below). All participants completed the task in an open-field environment. Like in Experiment 2, both navigation methods (*directions* and *states*) were available throughout the task, and participants completed 4 blocks of 6 trials, each with a different number of landmarks (2, 4, 8, or 16). Before the start of the main task, participants underwent a practice run of the map reading and navigation phases that was identical to the practice run in Experiment 2.

**Memory probe phase.** An additional feature of Experiment 3 were memory probe trials (S1F Fig). After the navigation phase on two randomly chosen trials on each block, participants experienced 'memory probes' in which they were asked to recall the location of two randomly chosen 'landmark' objects and the goal object. The location of these three objects were probed in random order.

Participants saw a 10 × 10 grid display (an 8 × 8 interior grid with visible boulders representing the surrounding walls) representing a bird's eye view of the environment, analogous to the display they had seen in the map reading phase. Participants were asked to recall the location of the probed object. The image of the currently probed object was displayed at the top of the screen. Participants were required to click on the grid square that they thought contained the probed object.

Participants were awarded additional points based on how close they were to the true landmark or goal location. They were awarded 30 additional points for placing the objects at the correct location, 20 points for placing it one step away, 10 points for placing it two steps away, and no points for placing it more than two steps away. These points contributed to participants' performance-based bonus payments. After clicking on a square, the selected square would be highlighted grey for 1 s. After the grey square had turned white, the next probe trial would start, with the object at the screen changing to the one corresponding to the next probe.

Participants did not get feedback after each probe. Instead, participants viewed a feedback screen at the end of the memory probe phase indicating the total number of points they had earned on all three probes. Finishing the memory probe phase would lead participants to either the map reading phase of the next trial, or a screen indicating the end of the block if this were the last trial in the block.

In the practice run of the memory probe phase, they practiced responding to three memory probes from objects they had seen in the practice run of the map reading phase.

**Debriefing.** After completing the task, participants were asked to describe what strategies they used to complete the task and whether they had used memorization aids (e.g., writing things down) to complete the task. Participants were told that admitting to using memorization aids would not affect their compensation for completing the task. No participants admitted to using memorization aids in Experiment 3.

### Day 2

**Pre-task questionnaires.** Participants again provided informed consent through an online consent form. There were no questionnaires to fill on this day of the task. They re-read the instructions and completed the practice runs again.

**Conditions.** On Day 2 of the study, participants were split into the 'free sampling' and 'forced sampling' conditions. Participants in the 'forced sampling' condition experienced a task similar to Day 1 of the study. However, participants in the free-sampling condition were allowed to sample their own 'landmarks' in the learning phase. The navigation and memory probe phases in both conditions were identical to Day 1 of the study. The main modifications on Day 2 were to the map reading phase of the study.

**Free sampling condition—map reading phase.** In the free-sampling condition, the map reading phase was modified to allow participants to freely select their own landmarks (S1C Fig). Participants viewed the 10 × 10 schematic of the grid as in other versions of the experiment, but without any highlighted blue squares. They were given a budget of 16 clicks and could click anywhere in the grid to reveal the object associated with the grid location. After clicking on a square, the image of the object associated with that location would appear. This image would disappear once participants clicked on the next grid square, or after 3 s if participants had not clicked on another grid square by then. The remaining number of clicks was always displayed centrally at the bottom of the screen.

There were very few constraints as to how participants could allocate these clicks. While the landmark locations were not allowed to be on the edge of the grid in Experiment 1, Experiment 2, and day 1 of Experiment 3, participants were allowed to sample landmarks on the border of the grid if they wished to. Participants were allowed to re-sample the same grid location using their budget of 16 clicks. In other words, participants could choose to sample 16 different locations on the grid or to sample as few as 2 locations by re-clicking on the same two locations again and again. The only restriction to click allocation was that participants were not allowed to sample the same grid square consecutively. After participants had exhausted their budget of 16 clicks, a grid square highlighted yellow would appear after one second, and participants would be required to click on a square highlighted yellow to reveal the goal object for the trial. After clicking on the yellow square, the goal object would be revealed at the location, and the 'Done' button would appear after one second. The goal object remained on the screen until the 'Done' button was clicked or three seconds after it appeared, whichever occurred sooner.

In the practice run for the free sampling condition, participants practiced freely sampling their own landmark locations in the map reading phase of the practice run.

**Forced sampling condition—map reading phase.** The forced-sampling condition was designed as a control to the free-sampling condition. Participants were not allowed to sample their own landmarks. Instead, participants were again required to click on squares highlighted blue to reveal the objects associated with each state; note therefore that 'forced sampling' is a feature of experiments 1 and 2 above, and Day 1 of the present experiment 3.

However, the way these 'landmarks' were chosen on Day 2 was different from Day 1 of the experiment. The landmark sampling method was designed as a control to the free sampling condition, which had fewer constraints on where and how landmarks could be placed than on Day 1. On each trial, we uniformly sampled a random number of unique landmarks to be between 2 and 16 (inclusive). This was to reflect the fact that participants in the 'free sampling' condition could choose any number of unique landmarks that were not necessarily a factor of 16. We then allocated the 16 clicks

over randomly chosen landmark locations. Since participants in the free-sampling condition were allowed to choose landmarks on the edge of the grid, we also allowed these randomly chosen landmarks to be placed on the edge of the grid. After these 16 clicks, participants were required to click on a square highlighted yellow, which would reveal the goal object for the trial. Besides the difference in landmark selection, every other aspect of the map reading phase was identical to the map reading phase in Experiments 1 and 2, and Day 1 of Experiment 3.

**Navigation phase.** The navigation phase of the trials proceeded in the same way as Day 1, with one exception. While start locations were chosen to be non-landmark locations on Day 1, participants occasionally started on top of a landmark location on Day 2 of the experiment. This was due to a bug in the experiment code which affected both conditions on Day 2. Excluding trials in which participants started on a landmark state did not affect the conclusions of any of our behavioral analyses.

**Memory probe phase.** The memory probe phase proceeded in an identical fashion to Day 1 of the study. Two trials out of each block were randomly selected to include a memory probe phase. In the free sampling condition, the probed landmarks were selected from those freely sampled by the participants.

**Debriefing.** After completing the task, participants were asked to describe what strategies they used to complete the task, and participants in the 'free sampling' condition were asked whether they had chosen to re-sample squares and why they might have might or might not have felt the need to do so. Participants were also asked whether they had used memorization aids for the task. Participants were told that admitting to using memorization aids would not affect their compensation for completing the task. No participants admitted to using memorization aids in Experiment 3.

**Principal component analyses.** For the data in the free-sampling condition, we computed the total number of times each participant sampled each location across all blocks, giving us 64 values (one for each location) per participant. We performed Principal Component Analysis (PCA) on the resulting subjects * locations matrix to reduce all participants' sampling data into a low-dimensional space that explained variance in sampling strategies across participants (S5E Fig).

### Deep reinforcement learning models

**Agent architecture.** The agent consisted of an LSTM with 100 recurrent units. These units fed into separate policy and value heads, both of which consisted of two fully connected layers of 64 units each. We used a Tanh activation function for units in the policy and value heads. Invalid actions (i.e., actions that walk into walls or obstacles, or response types that are not available on the current step) were masked from the output of the policy head by replacing the output logits with large negative numbers during choice time to prevent invalid actions from being taken [75].

### Task environment.

**Task set-up.** We designed the task environment to resemble human experience of the task in Experiment 1. Agents navigated an 8 × 8 grid where every grid was represented by an 'object'. Instead of individual images, the identities of each state were represented using an integer encoding, with a unique integer from 1 to 64 representing the identity of each state in the grid. Just like how humans experienced different images in different locations every trial, the mapping from each integer to each grid location was shuffled on every trial, and the agent did not have access to the ground-truth mapping between each integer and each location. However, the presence of an obstacle or wall was always indicated by the integer 65.

The agents were trained in a randomly interleaved fashion on all four conditions of the task: 'both', 'directions only', 'states only', and 'random alternation'. The agent was also trained on both the open-field and cluttered environments in a randomly interleaved fashion. For deep RL agents (but not human participants), the trial was terminated early if agents did not reach the goal before 200 steps.

**Simulated spatial memory from map reading phase.** Unlike humans, the agents did not explicitly undergo a map reading phase. Instead, we simulated the map reading phase by assuming that agents entered the navigation phase with partial and imperfect knowledge of the grid. The agents had access to estimated *x*- and *y*-coordinates of landmarks and goals, with the noisiness of these estimates designed to match human errors on the memory probe trials in Experiment 3.

Agents were assumed to have a noisy estimate of the goal location. We trained the agent on various degrees of noise, simulating the diversity in memory abilities in human participants. On each trial, we drew a value for goal noise from a distribution designed to approximately resemble the distribution of human errors on memory probe trials for goals in Experiment 3. Specifically, the amount of goal trial noise on each trial was drawn from the distribution:

$$s_{goal} \sim N(0.91, \ 0.44) \tag{1}$$

We then determined the agent's estimate $x$ and $y$-coordinates for the goal added noise to the ground truth coordinates:

$$\text{Estimated x/y coordinate} = \text{Ground x/y coordinate} + \varepsilon, \text{ where } \varepsilon \sim N(0, \ s_{goal}) \tag{2}$$

Similarly, we assumed that agents had imperfect memory for landmark locations. On each trial, the environment would consist of 2, 4, 8, or 16 landmarks. In our human experiments, the number of exposures were kept the same no matter how many unique landmarks there were, such that 2 landmarks would be repeated 8 times each, 4 landmarks would be repeated 4 time each, etc. Participants hence had stronger memories for each landmark if the environment consisted of fewer, more well-learnt landmarks. We simulated this in agents by scaling the amount of noise with the number of 'exposures' experienced in the map-reading phase:

$$s_{landmarks} = m * \log(n_{exposures}) + c \tag{3}$$

where $n_{exposures}$ is the number of exposures to the landmark in the learning phase. To simulate the diversity of human memory abilities, we fit linear regression lines for each participant in Experiment 3 predicting their error on probe memory trials using the number of exposures they were exposed to the landmark in the learning trial. Then, to determine the agent's landmark noise, drew the slope $m$ and the $y$-intercept $c$ from normal distributions based approximately on the distribution of slopes and $y$-intercepts we observed in humans. The distributions we drew $m$ and $c$ from were:

$$m \sim N(-0.63, \ 0.34) \tag{4}$$

$$c \sim N(1.90, \ 0.60) \tag{5}$$

The estimated landmark $x$ and $y$-coordinates were hence determined by:

$$\text{Estimated x/y - coordinate} = \text{Ground truth x/y - coordinate} + \varepsilon, \text{ where } \varepsilon \sim N(0, \ s_{landmark}) \tag{6}$$

**Navigation phase.** The agents were trained the optimize for performance in the navigation phase. On every step, the agents would receive as input an input vector describing the current observable state of the environment and information from their 'spatial memory'. Then, it would choose an action based on the output logits of their policy heads. This could be one of four 'direction' actions or one of four 'state' actions, depending on which actions were available on the current trial.

The input vectors were of length 48. Firstly, the agents received input describing which task conditions they were in, and which types of actions were currently available. This included: a one-hot vector representing which action condition the agent was in, an indicator for whether *direction* responses were available on the current step, and an indicator for whether *state* responses were available on the current step. This was of length 6 in total. If agents were in a cluttered environment, they would receive as additional input the $x$- and $y$- coordinates of the obstacles. This vector was of length 16. If agents were in an open-field environment, this vector would be all zeros.

Secondly, the input vector included information about the identity of the current, adjacent, and goal states. This was analogous to the images of objects that humans observed on the screen—while state identities were represented by distinct images for human participants, agents represented each unique as a unique integer (as explained above). Hence, the agent would receive as input a six-integer vector: the identity of the current state, the four adjacent states, and the goal state.

Moreover, the input vector included information drawn from the agents' simulated spatial memory. We assumed that humans attempted to recall the locations of states that appeared on the screen. Hence, we gave as input to the agents estimated $x$- and $y$- coordinates of all the states corresponding to the six 'integers' in the abovementioned state identity vector (i.e., states that would have been visible on screen to humans). This included: the agents' estimated $x$- and $y$-coordinates of the current state (or zeros if the current state was not a landmark), the estimated $x$- and $y$-coordinates of the four adjacent states (or zeros if the adjacent state is not a landmark) and the estimated $x$- and $y$-coordinates of the goal state. In total, the vector of the agents' estimated coordinates was of length 12. On top of this, we also assumed that humans had some insight into how well they had remembered goal and landmark locations. Hence, we included as additional input for the agent the amount of noise ($s_{goal}$ and $s_{landmark}$) that was added to the estimated coordinates of the current state, adjacent states, and goal state. This 'noise' vector was of length 6.

Lastly, when human participants made a step in the task, they would observe an animation indicating which direction they are moving regardless of whether they made a move using the *directions* or the *states*. We also include this directional information for agents by including as additional inputs the displacement in $x$ and the displacement in $y$ that resulted from the agent's last action. For example, if the agent moved left on the last step, the displacement in $x$ would be −1 and the displacement in $y$ would be 0. This vector was hence of length 2.

The table below (Table 1) summarizes the input vector received by the agent on every step:

Table 1. Summary of the input vector received by deep RL agents, showing the information provided to the agents and the length of each information encoded.

| Input information | Length |
| --- | --- |
| Action condition (i.e., both, directions only, states only, or random alternation) as a one-hot vector | 4 |
| $X$- and $y$-coordinates of each of the 8 obstacles | 16 |
| Whether state actions are currently available | 1 |
| Whether direction actions are currently available | 1 |
| Identity of the current state as an integer | 1 |
| Identities of each of the 4 adjacent states as integers | 4 |
| Identity of the goal state as an integer | 1 |
| Estimated $x$- and $y$-coordinates of the current state | 2 |
| Estimated $x$- and $y$-coordinates of each of the 4 adjacent states | 8 |
| Estimated $x$- and $y$-coordinates of the goal state | 2 |
| Estimated confidence in the coordinates of the current state | 1 |
| Estimated confidence in the coordinates of the 4 adjacent states | 4 |
| Estimated confidence in the coordinates of the goal state | 1 |
| Displacement in the previous step in the $x$- and $y$-directions | 2 |

## Training algorithm

The algorithm used for training the deep reinforcement learning models was based on an implementation of Proximal Policy Optimization (PPO) in Stable Baselines 3 [76]. Specifically, we based our deep RL model off a custom implementation of the PPO algorithm that allowed for both recurrent architectures and maskable actions (implemented by ref. [77]). PPO is a policy gradient RL algorithm that uses a clipped surrogate objective to prevent the policy from changing too quickly. The loss function consists of the policy loss, the value function loss, and the entropy loss:

$$L(\theta) \;=\; L^{\text{Policy}}(\theta) \;+\; c_1 L^{\text{Value}}(\theta) \;-\; c_2 L^{\text{Entropy}}(\theta) \tag{7}$$

The policy loss can be expressed as:

$$L^{\text{Policy}}(\theta) \;=\; \mathrm{E}_t[\min(r_t(\theta)\, A^t,\; \text{clip}(r_t(\theta),\, 1-\epsilon,\, 1+\epsilon)\, A^t)] \tag{8}$$

where $A^t$ is the advantage function (which measures how good or bad it is to take a specific action compared to the average action value at state $t$), $r_t(\theta)$ is the ratio between the new policy and old policy at state $t$, and $\epsilon$ is a hyperparameter that controls the amount of clipping. The value function loss is given by:

$$L^{\text{Value}}(\theta) \;=\; \mathrm{E}_t[(V\theta(s_t) - R_t)^2] \tag{9}$$

where $V_\theta(s_t)$ is the predicted value of state $s_t$ according to the value function parameterized by $\theta$ and $R_t$ is the observed overall returns from state $s_t$. Lastly, the entropy loss is given by:

$$L^{\text{Entropy}}(\theta) \;=\; -\mathrm{E}_t[\pi_\theta(a_t \mid s_t)\, \log \pi_\theta(a_t \mid s_t)] \tag{10}$$

where $\pi_\theta(a_t \mid s_t)$ is the probability of taking $a_t$ at state $s_t$ according to the policy parametrized by $\theta$. The policy and entropy losses are calculated only for actions that are allowed and not masked.

The following table (Table 2) shows the hyperparameters used for training:

To speed up training, we vectorized the task environment so that we could train the agent on 128 instances of the task environment in parallel. The agent's weights were updated every 8,192 steps (i.e., batches of 1,048,576 steps in total across the 128 environments). We trained the agent for 8,000 batch updates. We trained 20 iterations of the model, each initialized with a different random seed.

**Model behavior.** To evaluate the models' performance on the task, we tested each of the 20 models on a total of 4,000 trials each, which included 1,000 trials in each action condition (i.e., whether both 'direction' and 'state' actions were allowed, only either type of action was allowed, or a random type of action on each step was allowed). There were also an equal number of trials whether obstacles were present and absent, and for each number of landmarks (2, 4, 8, or 16). Like during training, invalid actions (i.e., actions that bring the agents towards an obstacle or that are not allowed

**Table 2. Hyperparameters used for model training.**

| Hyperparameter | Value |
| --- | --- |
| Learning rate | 0.0003 |
| Value function coefficient ($c_1$) | 0.5 |
| Entropy loss coefficient ($c_2$) | 0.001 |
| Clipping range ($\epsilon$) | 0.2 |
| Gradient clipping max norm | 0.5 |

under the current action condition) were masked. Unlike during training, agents chose actions deterministically rather than stochastically (i.e., always choosing the action with the highest probability).

**Classification of unit response patterns.** To classify LSTM units into 'landmark', 'spatial', or 'conjunctive' cells, we tested each agent on four different environments, each with two landmarks at different locations ((2, 2) and (5, 5), (2, 5) and (5, 2), (2, 2) and (5, 2), and (2, 5) and (5, 5)). Each model was tested on 1,000 trials for each environment. We computed each unit's mean cell state response at each of the 64 locations in each environment, resulting in 256 data points for each unit. For each model, we smoothed the 2-dimensional location matrix using a Gaussian filter (with standard deviation $s = 0.7$). We then conducted an ANOVA examining how the mean cell state response in each location in each environment varied according to two factors: the quadrant each location belonged to (i.e., bottom left, bottom right, top left, or top right) and whether the location was on or adjacent to a landmark in each environment. We classified units as 'spatial' units if there was only a significant main effect of quadrants, as 'landmark' units if there was only a significant main effect of landmarks, or as 'conjunctive' units if both main effects were significant (i.e., suggesting an additive effect of spatial position and landmark presence), and/or if there was a significant interaction effect (i.e., suggesting a multiplicative effect of spatial position and landmark presence).

**Identification of functional modules.** To classify LSTM units into functional 'modules', we regressed the cell state responses of each LSTM unit onto each of the 8 action logits from the output of the policy network and averaged the $R^2$ values for regressions involving either the 4 'direction' actions or 4 'state' actions. We classified the 10 units that explained the most variance in 'direction' actions as 'vector' units and the 10 units that explained the most variance in 'state' actions as 'transition' units. As a control for subsequent analyses, we also identified the 10 units that explained the least variance in both 'direction' and 'state' actions as 'unspecialized' units.

**Lesion analysis.** To lesion the 'vector', 'transition', and 'unspecialized' modules for each model, we zeroed out the cell and hidden state activations for units in each module. We then tested the lesioned networks on the *'both'*, *'directions only'* or *'states only'* versions of the task in both open field and cluttered environments. For each lesioned model, we tested the model on 4,000 trials in total: 2000 each in the *'directions only'* and *'states only'* conditions, and 2000 each in the open field and cluttered environments with all conditions counterbalanced.

**Decoding analyses.** For each model, we attempted to decode the current and goal locations from the cell state responses 'vector', 'transition', and 'unspecialized' units. To do this, we trained Ridge regression models (alpha = 0.5) to predict the x- or y-coordinates for the current or goal locations from the cell state activations of the 20 'vector', 'transition', or 'unspecialized' units. We trained the models on 80% and tested the models on a held-out 20% of the dataset. We evaluated the models' decoding errors by calculating the Euclidean distance between the predicted and the actual goal or current locations.

For each model, we also attempted to decode whether agents were adjacent to a goal or a landmark from each module. To do this, we trained logistic regression models to predict whether agents were adjacent to a goal or a landmark at each timestep using the cell state activations of the 20 'vector', 'transition', or 'unspecialized' units. We trained the models on 80% and tested the models on a held-out 20% of the dataset. The number of timepoints where models were adjacent to the goal (or landmark) were far outweighed by the timepoints where models were not adjacent to a goal (or landmark), we matched the number of samples in each class by under-sampling the number of samples in the majority (not adjacent to landmark/goal) class. We evaluated the models' decoding errors by calculating the proportion of held-out timesteps where the models misclassified whether the agents were adjacent to a goal or a landmark.

**Representational geometry.** For a cleaner visualization of the model's representational geometry, we ran 80,000 trials of one representative model on the 'both' condition of the task. We performed a PCA on the cell state activations of the 'vector' and 'transition' units and computed the mean PCs when the agent was at each location in the grid under each condition (i.e., before or after a landmark was encountered, at a landmark or non-landmark).

**Testing on human-selected landmarks.** Experiment 3 involved different constraints for selecting landmarks: unlike Experiments 1 and 2, landmarks were allowed to be selected on the edge, and the number of samples need not be evenly distributed across all unique landmarks (i.e., the 16 landmarks could include 8 samples of one landmark, 6 samples of another, and 2 samples of another). This was true for both the free-sampling and forced-sampling conditions on Day 2: i.e., humans were free to select landmarks without the same constraints as in Experiment 1 and 2, and we therefore designed the control forced-sampling condition to involve randomly selected landmarks without these constraints as well. Because our earlier models were trained on landmarks selected with these constraints, we had to retrain a model on landmarks that did not have these constraints. Training followed the exact same training regime as described above, only with landmarks randomly selected as in the forced-sampling condition on Day 2 of Experiment 3 (see above).

To test whether the model performed better on human-selected landmarks, we 'simulated' each participant in Experiment 3 by testing the model on environments with the landmark configurations that each participant experienced on Day 2. This meant that for participants in the free-sampling condition, the model was tested on landmark configurations chosen by the participants themselves, but in the forced-sampling condition, the model was tested on random landmark configurations that the participants experienced. The model's 'memory' of landmarks for each simulated participant was also determined by their performance on memory probe trials, such that the slope and intercept ($m$ and $c$ in (Equation 3) were fit to each individual participants' pattern of errors on memory probe trials.

## Mixed effects models

All mixed effects models, except the one for number of landmarks, were fit using the *MixedModels* package in Julia [78]. Model selection for all mixed effects models begun with using the maximal model (i.e., with random intercepts and slopes for all predictors). We then identified potential overparmeterization and selected a parsimonious model by reducing the model complexity until a further reduction implies a significant loss in the goodness-of-fit as measured by with Likelihood Ratio Tests with $a = 0.2$ (as recommended in ref. [79]). However, for our main results, the pattern of significant results produced by the reduced models were not different from the results from the maximal model with random slopes and intercepts (i.e., the most conservative model). The models for the number of landmarks (Fig 2E) were fit using the *brms* package in *R* to allow for Bayesian model selection.

## Supporting information

**S1 Fig. A: Schematic diagram of the map reading phase in Experiments 1 and 2, Day 1 of Experiment 3, and the forced sampling condition in Day 2 of Experiment 3. B:** Schematic diagram of the map reading phase in the free sampling condition in Day 2 of Experiment 3. **C:** Schematic diagram for the navigation phase. **D:** View when the *directions* display was disabled in Experiment 1. **E:** View when the *states* display was disabled in Experiment 1. **F:** Schematic diagram for the memory probe phase in Experiment 3.
(PDF)

**S2 Fig. Participants' (above) and models' (below) use of vector-based responses (*y*-axis) as a function of destination type (i.e., goal, landmark, or non-landmark; *x*-axis) and whether the state had been visited before (color of bar), split by whether navigation was in an open field (left) or cluttered (right) environment.** Data underlying this figure is available at https://osf.io/w39d5/.
(PDF)

**S3 Fig. A/B: Relationship between the proportion of steps made using vector-based responses (*x*-axis) and the number of steps taken to reach a goal (*y*-axis; represented on a logarithmic scale) for each participant in Experiment 2 (A) and Experiment 3 (B).** Each point represents a single person, while the lines represent the best-fitting

quadratic curve. **C/D:** Participants' use of vector-based responses (*y*-axis) as a function of destination type (i.e., goal, landmark, or non-landmark; *x*-axis) and whether the state had been visited before (color of bar) in Experiment 2 **(C)** and Experiment 3 **(D)**. Data underlying this figure is available at https://osf.io/w39d5/.
(PDF)

**S4 Fig. Relationship between scores on the Navigation Strategy Questionnaire (NSQ; *x*-axis) and performance on the task (*y*-axis), as measured by accuracy (left) or number of steps to goal (right; presented on a logarithmic scale).** Each dot represents and individual participant and the line represents the best-fitting line from a linear regression. Data underlying this figure is available at https://osf.io/w39d5/.
(PDF)

**S5 Fig. A: Mean error (in Euclidean distance) on memory probe trials for landmarks in each of the landmark number conditions.** Each dot represents and individual participant and error bars represent the 95% CI. **B:** accuracy in navigation (*y*-axis) across the number of landmark conditions (*x*-axis) before and after a landmark is encountered (hue). Each dot represents an individual participant and error bars represent the 95% CI. **C:** Proportion of time spent across the whole experiment before encountering a landmark (*y*-axis) across the number of landmark conditions (*x*-axis). Each dot represents and individual participant and error bars represent the 95% CI. **D:** Mean number of landmarks used (i.e., clicked on using a state-based response; *y*-axis) in the different number of landmark conditions. Each dot represents an individual participant and error bars represent the 95% CI. **E:** Accuracy in navigation (*y*-axis) across the number of landmark conditions (*x*-axis) before and after the edge of the grid is encountered (hue). Each dot represents an individual participant and error bars represent the 95% CI. **F:** Proportion of time spent on each trial spent on the edge of the grid (*y*-axis) across the number of landmark conditions (*x*-axis). Each dot represents an individual participant and error bars represent the 95% CI. Data underlying this figure is available at https://osf.io/w39d5/.
(PDF)

**S6 Fig. A: Percentage of steps where participants used a landmark (defined as clicking on the image of a landmark on the state-based display; *y*-axis) as a function of the order in which they passed by the landmark on a navigation trial.** Error bars represent the 95% CI. **B:** Percentage of steps where participants used a landmark (defined as clicking on the image of a landmark on the state-based display; *y*-axis) as a function of the order in which they passed the landmark on a navigation trial. Error bars represent the 95% CI. **C:** Same as **B** but focusing only on the first landmark participants pass by. **D:** Same as **B and C** but focusing only on subsequent landmarks that participants pass by. **E:** Percentage of all landmarks used in the navigation phase (*y*-axis) that were presented in each serial position in the map-reading phase. Each plot represents a condition with a different number of landmarks. When there were more than 16 landmarks, the same landmarks were presented several times: the lightest hue in each plot indicates the first time the landmarks were presented, with progressively darker hues indicating each subsequent presentation. Error bars represent the 95% CI. Data underlying this figure is available at https://osf.io/w39d5/.
(PDF)

**S7 Fig. Training curves for deep meta-RL agents showing how (A) mean episode length and (B) reward evolves with training epoch.** Each red line represents the training curve for a single model with a single random seed, while the black line represents the mean across all models.
(PDF)

**S8 Fig. A: Performance as measured by number of steps taken to goal (*y*-axis; presented on a logarithmic scale) as a function of number of landmarks (*x*-axis) for models with noisy landmark and goal encodings.** Each dot represents a model with a different initialization seed, and the error bars represent the 95% CI. **B:** Same as **A** but for models with perfect memory of landmarks and goals. **C:** Models' use of vector-based responses (*y*-axis) as a function

 

of destination type (i.e., goal, landmark, or non-landmark; *x*-axis) and whether the state had been visited before (color of bar). Each dot represents the behavior of an individual model, with error bars representing the 95% CI. **D:** Same as **C** but for models with perfect memory of landmarks and goals. **F:** Mean number of landmarks used (i.e., clicked on using a state-based response; *y*-axis) in the different number of landmark conditions. Each dot represents an individual model and error bars represent the 95% CI. **G:** Same as **C** but for models with perfect memory of landmarks and goals.
(PDF)

**S9 Fig. Stacked bar plots showing the proportion of response pattern types ('stable spatial', 'remapping spatial', or 'non-spatial') across all units or in the 'vector', 'transition', or 'unspecialized' clusters.** 'Vector' units are more likely to have stable spatial responses, while 'transition' units are more likely to have remapping spatial responses.
(PDF)

**S10 Fig. Sample of activation heatmaps (left) and spatial autocorrelograms (right) for units with the highest 'gridness' scores.** 'Gridness' scores are shown on top of each unit's spatial autocorrelogram.
(PDF)

**S11 Fig. First three principal components for the PCA on the cell state activations of 'vector' units (A and D), 'transition' units (B and E), and 'unspecialized' units (C and F).** Each dot represents the centroid of the PCs for each location in the grid. In the left plots **(A, B, and C)**, red and blue dots represent the PCs before and after a landmark is encountered, respectively. In the right plots **(D, E, and F)**, green and purple dots represent the PCs at a landmark or a non-landmark. PCA results are only shown for one representative model.
(PDF)

**S12 Fig. A: First three principal components for the PCA on the cell state activations of 'vector' units.** Each dot represents the centroid of the PCs for each location in the grid. Red and blue dots represent the PCs before and after a landmark is encountered, respectively. PCA results are only shown for one representative model. **B:** First three principal components for the PCA on the cell state activations of 'transition' units. Each dot represents the centroid of the PCs for each location in the grid. Green and purple dots represent the PCs for non-landmarks and landmarks, respectively. The representations of 'transition' units seem to separate landmarks and non-landmarks without apparent spatial structure. Note that the edges are not represented for landmarks because the task did not allow landmarks to be placed on the edges.
(PDF)

**S13 Fig. A: Plot showing the variance explained by each exploration component. B: Graph illustrating four different configurations of four landmarks and their relationship with the mean distance between all locations and their nearest landmark** . The two configurations of four landmarks that appear to be represented by the second exploration principal component minimize the mean distance to the nearest landmark. **C:** Correlation plots representing the relationship between the three exploration-related principal components and the three hypothesis-driven metrics (mean distance to nearest landmark, mean distance to center, and mean error on probe trials). In the plots for the mean distance to the nearest landmark (first column), the hues represent a median split in the mean number of unique landmarks chosen by participants. Blue represents participants who chose a lower number of unique landmarks, while orange represents participants who chose a higher number of unique landmarks.
(PDF)

## Acknowledgments

We are grateful to Leonie Glitz for helpful discussions and valuable input which contributed to the conceptual development of this work.

## Author contributions

**Conceptualization:** Denis C. L. Lan, Laurence T. Hunt, Christopher Summerfield.

**Data curation:** Denis C. L. Lan.

**Formal analysis:** Denis C. L. Lan.

**Funding acquisition:** Christopher Summerfield.

**Investigation:** Denis C. L. Lan.

**Methodology:** Denis C. L. Lan, Laurence T. Hunt, Christopher Summerfield.

**Supervision:** Laurence T. Hunt, Christopher Summerfield.

**Validation:** Laurence T. Hunt, Christopher Summerfield.

**Visualization:** Denis C. L. Lan.

**Writing – original draft:** Denis C. L. Lan, Christopher Summerfield.

**Writing – review & editing:** Denis C. L. Lan, Laurence T. Hunt, Christopher Summerfield.

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
