## [Editor Report · Decision Letter 0]

Dear Dr Lan,

Thank you for submitting your manuscript entitled "Goal-directed navigation strategies in humans and deep meta-learning agents" for consideration as a Research Article by PLOS Biology.

Your manuscript has now been evaluated by the PLOS Biology editorial staff as well as by an academic editor with relevant expertise and I am writing to let you know that we would like to send your submission out for external peer review.

Once your full submission is complete, your paper will undergo a series of checks in preparation for peer review. After your manuscript has passed the checks it will be sent out for review. To provide the metadata for your submission, please Login to Editorial Manager (https://www.editorialmanager.com/pbiology) within two weeks, i.e. by Jan 02 2025 11:59PM.

Kind regards,

Christian

Christian Schnell, PhD

Senior Editor

PLOS Biology

cschnell@plos.org

---

## [Decision Letter · Decision Letter 1]

Dear Dr Lan,

Thank you for your patience while your manuscript "Goal-directed navigation strategies in humans and deep meta-learning agents" was peer-reviewed at PLOS Biology. It has now been evaluated by the PLOS Biology editors, an Academic Editor with relevant expertise, and by several independent reviewers.

In light of the reviews, which you will find at the end of this email, we would like to invite you to revise the work to thoroughly address the reviewers' reports.

As you will see below, the reviewers are overall supportive of your study, but Reviewer 1 is concerned that the findings are limited to this particular task. Reviewer 2 mainly asks for textual revisions, while Reviewer 3 has a number of specific questions for some of the experiments that will likely require additional analyses.

Given the extent of revision needed, we cannot make a decision about publication until we have seen the revised manuscript and your response to the reviewers' comments. Your revised manuscript is likely to be sent for further evaluation by all or a subset of the reviewers.

**IMPORTANT - SUBMITTING YOUR REVISION**

*Re-submission Checklist*

*Published Peer Review*

*PLOS Data Policy*

*Blot and Gel Data Policy*

Sincerely,

Christian

Christian Schnell, PhD

Senior Editor

PLOS Biology

cschnell@plos.org

REVIEWS:

Reviewer #1: The authors present the results of three online behavioural studies that examine the strategies used to support 'few-shot' goal-directed navigation in novel environments. Specifically, they contrast the use of topological / transition-based strategies and geometric / vector-based strategies using a task in which participants have to navigate around an 8 x 8 grid, wherein each location is either marked by a unique visual 'landmark' or contains an obstacle. Participants begin by seeing an overhead view of the map without any visible landmarks, and are prompted to click on between 2 and 16 locations to reveal the visual landmark therein, before being prompted to click on the goal location. They are then placed at a random start location and must choose between two navigation strategies for each move - either selecting a visual landmark to move to (from those immediately adjacent to their current location), or a movement direction to move in.

In Experiment 1, the authors demonstrate that performance is best when participants can make use of either strategy, rather than being limited to one or the other on each turn; that a vector based strategy is less useful in cluttered environments; and that participants who tended to rely on a single strategy (when both were available) performed poorly. In Experiment 2, the authors demonstrate that the number of landmarks viewed prior to navigation has no effect on performance; and that transition and vector based strategies are preferentially associated with landmark / conjunctive and stable spatial responses in a deep learning agent, respectively. In Experiment 3, participants were given the option to choose which landmarks to view prior to navigation, which led to better performance. This was driven by selecting landmarks that were well distributed across the environment and close to the centre.

The manuscript is clearly written, the experimental task is novel, the analyses are thorough, and these results may be of interest to the field. My only concern is what we can conclude from these findings - i.e. what do they tell us about real world navigation? My concern is that this stylised task bears little resemblance to real world navigation, and so it is not clear what can be concluded or extrapolated from these findings. These concerns are described in more detail below. Given the content, I also wonder if this manuscript is suitable for publication in PLoS Biology? Would these results not be more suitable for PLoS Computational Biology, given the focus on comprehensive analysis of behaviour and deep meta-RL models?

Major Concerns

It is not immediately clear what we can conclude from these findings, because I am not sure how ethologically valid this task is. Is seems odd to be unaware of one's starting location when navigating, and similarly, to be unaware of the relative location (or direction) of landmarks as you pass them. I find it worrying that the number of landmarks viewed before navigation began had no impact on performance (Fig 2D). Overall, my concern is that the authors have very accurately characterised performance on this task, but told us little about navigational strategies in the real world. The authors allude to this possibility in the Discussion. Can they point to anything in the data that validates these findings? For example, did performance correlate with scores on either of the questionnaires that participants completed?

The task could be explained more clearly in the main text, as this is crucial to understand the results that follow. Specifically, it would be useful to mention that: (i) during the map reading phase, participants have a single forced choice of which (blue) square to click on, and these landmark images remain on screen for <3s; (ii) start locations were always four steps away from the goal location; and that (iii) one of the adjacent locations could be marked with a boulder (i.e. obstacle), in which case clicking on this image (or the corresponding direction) would have no effect. It would also be useful to clarify whether there were any repeats of landmark stimuli across trials ("Every trial involved a grid with completely different sets of images…")?

Minor Comments

Several references are repeated (i.e. 8 and 9, 10 and 32, 24 and 29, 27 and 30 etc)

It is confusing to switch between 'obstacles' and 'clutter locations' - the use of a single term would be clearer

Figure 2D: It seems odd to me that participants would ever use a directional strategy to reach the goal when they are adjacent to it (and it can therefore be seen as one of the four landmark options) - can the authors offer any explanation of why this might be?

Page 13 / 14: Typo - "in favour for the null hypothesis"

Figure 4: The caption for panels B and D are reversed

Pages 20 / 23: Typo - 'lik3ly'

Discussion: Typo - "people rely predominantly on vector-based strategies to heading in the general goal direction"

Reviewer #2 (William de Cothi): This is a thorough, elegant and well-written study investing goal-directed navigation strategies in humans and comparing them to reinforcement learning agents implementing meta learning via the recurrent connectivity in an LSTM layer (e.g. once learnt, network weights are fixed). The key benefit to this type of comparison is that the network can learn to approximate optimal behaviour on complex tasks, enabling the authors to probe both the optimality of human behaviour and the mechanisms behind it. In sum, I find this study uses sound and novel methods to provide useful insights into the mechanisms driving human goal-directed behaviour, and the preregistration of their experiments is to be applauded.

I have some suggestions to the authors:

I appreciate the authors attempt to dissociate direction- and state-based strategies which is not trivial to achieve. However, in the real-world these two things are not so distinct. The deterministic nature of (state,action) -> next_state transitions in navigating physical space means you can just as easily characterise the same policy as a sequence of actions (directions) or the sequence of states they lead to (which the authors refer to transitions). Technically, these both describe the same sequence of transitions, just one is in a state-based format and the other in a direction based format, so I take issue with authors' use of the term 'transitions' for meaning 'state-based'. The format in which participants choose to implement that policy will likely vary depending on state quantity and saliency - remembering every state is difficult and high value states are more memorable, meaning participants may choose to encode parts of the policy through low-value states via directions, and parts of the policy through high-value states via the states themselves. I would suggest:

1. The author's use the term 'state-based' rather than 'transition-based'

2. The authors include something in their discussion on how these policies may not be so distinct in reality - I was actually expecting this to come in the paragraph starting line 712 so I would suggest it goes there

3. In this part of discussion the authors also discuss potential goal-directed strategies not considered in their study/analysis - for example how does this fit with 'beaconing' and 'dead-reckoning' strategies?

In the analysis of LSTM units, 'spatial' cells are stated as being classified depending on their modulation in activity over quadrants of the environment. Whilst this would potentially fit a 'hippocampal place cell' view of spatial cells (although I'm not sure it would if the place cell was centred on the intersection of all quadrants?), my understanding is it would exclude an entorhinal grid cell as being spatial (e.g. a cell with spatially stable firing fields distributed across the environments)? Indeed, its hypothesised grid cells in entorhinal cortex play an important role in the neural computation of vectors (Bush et al., 2015; Banino et al., 2018). I would suggest:

4. If the above is correct, the authors use spatial stability within an environment as a criteria for categorising spatial cells e.g. correlation between activation rate maps constructed from, for e.g., 1st half vs 2nd half of trials, or alternating trials, or for alternating time steps.

Some more minor points:

5. I agree with the the authors that one of the reasons to compare with deep RL agents is to form a comparison to a near-optimal strategy - therefore it would be helpful for them to should show that the networks' learning has plateaued during training, and is indeed near-optimal

6. Line 457 - the cells they describe here also resemble boundary cells found in mammalian subiculum (e.g. Lever et al., 2009)

Reviewer #3: This study investigates the arbitration between direction-based choice behavior and transition-based choice behavior in the context of spatial navigation in open vs cluttered environments. Human behavior is analyzed and compared to the action policies of trained RL agents. Core conclusions include 1. that humans do better when endowed with the possibility to use both behavioral strategies, 2. humans tend to use a transition-based strategy close to landmarks/goals. Below is elaborated several conceptual queries aimed at helping to deepen our understanding of these results and relate them to the literature.

One possible alternative to using landmarks is using the wall or boundaries. There is not any quantification of this behavior. For example, in experiment 2 do people use the wall more if there are fewer landmarks? Could this explain the flat curve for number of landmarks presented vs number of steps taken to goal?

Experiment 2: What was the average number of landmarks used in a single trial? If it is around 2 then maybe 2 landmarks is all you really need and this can explain the data. I saw percentages of directions vs state action choice but not a clear average number of different landmarks used.

Experiment 2: The RL models use different amounts of noise to model different numbers of landmarks presented. If memory is a possible explanation for the same performance across different number of landmarks presented, are the landmarks presented last more likely to be used?

Related to this, how does the agent behavior change if you make the goal/landmark encoding non-noisy? What are the predictions for this under the hypothesis that noisily encoding the landmarks/goals corresponds to a memory constraint.

Experiment 2: Is there a pattern to which landmarks get used? Are landmarks closer to the goal more likely to get used? Are the landmarks that are first encountered more likely to be used (in order to reduce the uncertainty due to unknown starting location)?

RL agent experiment: The RL agents were run on different numbers of presented landmarks. What does the plot recreating Figure 2E of number of steps to goal vs number of landmarks presented look like?

In general, its not clear why do the artificial agents develop transition vs state-based behavioral arbitration strategies? What ablations can be done to pinpoint the source of this? Its not clear why the artificial agents would care at all about transition vs state-based encoding. I find it a bit unconvincing to say that these agents are "approximately optimal" without a deeper interrogation into why they exhibit the pattern of action selection that they do.

Figure 4 displays some unit responses in the trained network. It is striking that no grid cell-like responses are observed apparently. I think this would be important to discuss. For example, what are the differences in this agent architecture (vs e.g. Banino et al, Cueva/Wei) that lead to a lack of "grid cells". Have you tried applying spatial periodicity or "gridness" scoring to the unit responses?

A prominent theme in spatial cognition is the use of neural representations for structural generalization. For example, if two environments had roughly the same structure, then the same cell population might be activated while navigating in both environments. Is there any evidence of this in the RL agents?

Presentation/minor comments:

What are the error bars in figure 2D? Please add info to caption.

Figure 1A and B. Might be clearer if the same map was used. You see the underlying structure that the participants dont see, then you see what the participants do see.

Figure 4 labels are mixed up from caption.

Figure 5E caption: could be clearer that these are the principal components of the samples in the 8x8 environment

Line 551 lik3ly

Line 670 - missing "work"

---

## [Decision Letter · Decision Letter 2]

Dear Dr Lan,

Thank you for your patience while we considered your revised manuscript "Goal-directed navigation strategies in humans and deep meta-learning agents" for publication as a Research Article at PLOS Biology. This revised version of your manuscript has been evaluated by the PLOS Biology editors, the Academic Editor and one of the original reviewers.

Based on the reviews and on our Academic Editor's assessment of your revision, we are likely to accept this manuscript for publication, provided you satisfactorily address the remaining points raised by the reviewers. Please also make sure to address the following data and other policy-related requests:

* We would like to suggest a different title to improve its accessibility for our broad audience. Which of the following would work best in your opinion?

"Both humans and deep neural networks rely on an adaptive mix of vector-based and landmark-based strategies to support goal-directed navigation"

OR

"Optimal human navigation in novel environments relies on an adaptive mix of vector-based and landmark-based strategies"

I'd be happy to discuss alternative titles, please send them to me via email before resubmitting your revised manuscript.

* Please add the links to the funding agencies in the Financial Disclosure statement in the manuscript details.

* Please include information in the Methods section whether the study has been conducted according to the principles expressed in the Declaration of Helsinki.

* DATA POLICY:

Regardless of the method selected, please ensure that you provide the individual numerical values that underlie the summary data displayed in the following figure panels as they are essential for readers to assess your analysis and to reproduce it: 2ADE, 3BCE, 4DEFG, 5ABCDF and similar panels in the supplementary figures.

* CODE POLICY

* If you have any references in the supplementary information, please move them to the main reference list.

* If you have further methodological details in the supplementary information, please move those to the main manuscript file as well.

We expect to receive your revised manuscript within two weeks.

*Published Peer Review History*

*Press*

Sincerely,

Christian

Christian Schnell, PhD

Senior Editor

cschnell@plos.org

PLOS Biology

Reviewer remarks:

Reviewer #3: I thank the authors for this thoughtful revision. Overall, the experimental design is clean, and the range of behavioral and model analyses is impressive. The manuscript clearly represents a significant amount of work and will be of interest to many researchers studying navigation, decision-making, and artificial agents.

That said, I still find myself seeking greater theoretical clarity around some of the core claims, particularly regarding the normative principles underlying arbitration between strategies. While the authors state that "agents' behavioural policies for arbitrating between direction- and state-based responses likely resulted from the demands of our task," and that these responses "place different representational and computational demands," the explanation remains somewhat opaque. Without a stronger normative or computational framing, it's difficult to evaluate what "optimal" arbitration actually means here—or whether the performance of DRL agents can be considered optimal in a meaningful sense. Potentially, the authors could soften the language surrounding "optimality" of arbitration and/or contextualize more specifically with respect to DRL agents (which may or may not be ``optimal'').

On the grid-like representation issue, I agree these are not grid-like units though some appear close to superpositions of plane waves. However, I'm unsure about the decision to largely set grid codes aside, especially since there is such a strong emphasis on the emergence of "representational geometry" from the "mammalian navigational system" in the DRL agents and the fact that, functionally, the authors link direction-based choice to vector-based navigation—a strategy closely tied to grid codes in the brain (e.g. Bush et al, Neuron 2015). Overall, the study is rigorous and thoughtful, however its focus on behavioral comparisons and the analysis of internal representations in artificial agents which do not exhibit key, and functionally relevant, spatial codes observed in biological organisms creates confusion for me regarding what the biological implications of this study are.

The comments about training objectives and input structure are helpful, though, to be clear, other models (e.g., Sorscher et al, Tolman-Eichenbaum) have recovered grid-like codes without velocity input per se. The analogous strategy to Banino et al would be to pretrain supervisedly on position and then train using RL "on top" of the presumably resulting grid-like populations. Then, one could see if the DRL agents "naturally choose" to use these grid codes for direction-base navigation or not.

The section on structural generalization is promising but would benefit from more further details — specifically, what kinds of environments generalize well and where generalization breaks down. However, I wonder if there is sufficient systematic diversity on environment structure to examine this and I understand the authors may prefer to leave this aside as an issue not core to their overall goal in this study.

---

## [Editor Report · Decision Letter 3]

Dear Denis,

Thank you for the submission of your revised Research Article "Goal-directed navigation in humans and deep reinforcement learning agents relies on an adaptive mix of vector-based and transition-based strategies" for publication in PLOS Biology. On behalf of my colleagues and the Academic Editor, Raphael Kaplan, I am pleased to say that we can in principle accept your manuscript for publication, provided you address any remaining formatting and reporting issues. These will be detailed in an email you should receive within 2-3 business days from our colleagues in the journal operations team; no action is required from you until then. Please note that we will not be able to formally accept your manuscript and schedule it for publication until you have completed any requested changes.

PRESS

Sincerely, 

Christian

Christian Schnell, PhD

Senior Editor

PLOS Biology

cschnell@plos.org